# Don't be lazy: CompleteP enables compute-efficient deep transformers

**Nolan Dey**[*]
Cerebras Systems

**Bin Claire Zhang**[*]
Cerebras Systems

**Lorenzo Noci**
ETH Zurich
Princeton University

**Mufan Li**
Princeton University

**Blake Bordelon**
Harvard University

**Shane Bergsma**
Cerebras Systems

**Cengiz Pehlevan**
Harvard University
Kempner Institute

**Boris Hanin**
Princeton University

**Joel Hestness**
Cerebras Systems

## Abstract

We study compute efficiency of LLM training when using different *parameterizations*, i.e., rules for adjusting model and optimizer hyperparameters (HPs) as model size changes. Some parameterizations fail to *transfer* optimal base HPs (such as learning rate) across changes in model depth, requiring practitioners to either re-tune these HPs as they scale up (expensive), or accept sub-optimal training when re-tuning is prohibitive. Even when they achieve HP transfer, we develop theory to show parameterizations may still exist in the *lazy learning* regime where layers learn only features close to their linearization, preventing effective use of depth and nonlinearity. Finally, we identify and adopt the parameterization we call *CompleteP* that achieves both depth-wise HP transfer and non-lazy learning in all layers. CompleteP enables a wider range of model width/depth ratios to remain compute-efficient, unlocking shapes better suited for different hardware settings and operational contexts. Moreover, CompleteP enables 12-34% compute efficiency improvements over the prior state-of-the-art. All experiments were run on Cerebras CS-3 systems. A minimal implementation is available at https://github.com/EleutherAI/nanoGPT-mup/tree/completep.

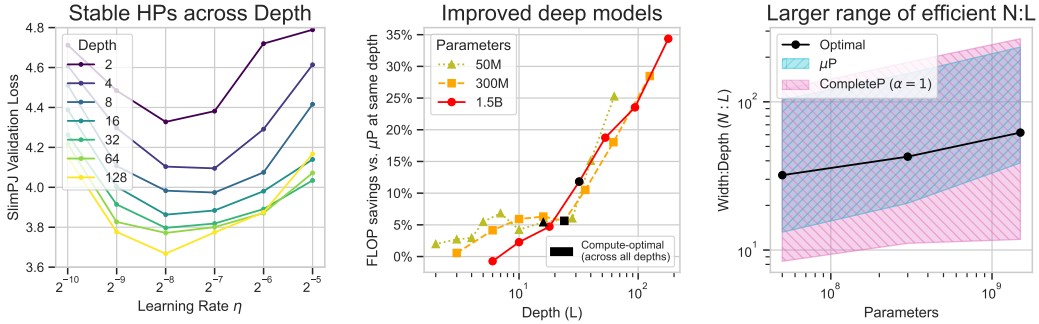

Figure 1: We introduce **CompleteP**, which offers depth-wise HP transfer (**Left**), FLOP savings when training deep models (**Middle**), and a larger range of compute-efficient width/depth ratios (**Right**).

[*]Equal contribution. Email: {nolan,claire.zhang,joel}@cerebras.net

39th Conference on Neural Information Processing Systems (NeurIPS 2025).

# 1 Introduction

A hallmark of modern deep learning is that training larger models leads to better performance [1]. In Large Language Models (LLMs), for instance, the paradigm of pre-training larger models on larger datasets has led to remarkable results in a wide range of downstream evaluations. However, these gains come at a substantial increase in computational cost. As compute budgets grow, practitioners must navigate a complex design space to choose model width and depth, dataset size, batch size, number of training steps, and a variety of other hyperparameters (HPs) in order to find an optimal allocation of resources to minimize a pretraining objective, given a fixed compute budget [2, 3].

The prohibitive cost of naively conducting such a search in large models can result in suboptimal HPs and hence an inefficient use of computational resources. This motivated techniques such as the maximal update parameterization (μP), which ensures that optimal HPs remain approximately constant when scaling model *width* [4, 5] and enables a "tune small and train large" strategy.

We refine and thoroughly compare extensions of μP for simultaneously scaling *depth and width* [6–8]. These scaling strategies define *parameterizations* — sets of rules for how to scale model and optimizer hyperparameters with model size. The core difference between μP and these depth-aware refinements is how they re-scale the outputs of the transformer's [9] residual block (as a function of depth $L$) before the output gets added to the residual stream variables $h^\ell$:

$$\mathbf{h}^{\ell+1} = \mathbf{h}^\ell + L^{-\alpha}\,\mathcal{F}_\ell(\mathbf{h}^\ell)\,,\; \ell \in \{1, ..., L\}, \tag{1}$$

where $\mathcal{F}_\ell$ is the $\ell$'th residual block (for transformers, these are MLP and attention blocks). The depth-dependent re-scaling factor is governed by a single parameter $\alpha \in [0.5, 1]$.

Yang et al. [7] argue $\alpha = 0.5$ works best in practice and that HP transfer is not possible at any $\alpha$, while Bordelon et al. [8] find $\alpha = 1$ allows better learning as $L \to \infty$. Since $\alpha \in \{0.5, 1\}$ are the two most promising potential candidates for depth scaling [6–8], we compare these two values and find $\alpha = 1$ is consistently more compute-efficient through better HP transfer and faster pre-training (Figure 1). Setting $\alpha = 1$ gives a parameterization we call *CompleteP* because it is the unique $\alpha$ ensuring *complete feature learning* as width and depth are scaled (Section 6).

To realize these gains, however, we must extend the parameterization to include principled re-scalings of LayerNorm (LN) and bias learning rates, AdamW's weight decay $\lambda$, and AdamW's $\epsilon$ (Table 1). CompleteP is simple to implement, yet yields superior upstream and downstream performance, with gains over alternative approaches increasing with model depth. To summarize, our contributions are:

- **HP transfer across depth.** We compare transfer of learning rate and weight initialization standard deviation across depth (2-128 layers) for the standard parameterization (SP), μP, and $\alpha = \{0.5, 1\}$, when training Pre-LN transformers (Figure 2, 3). We find only $\alpha = 1$ enables depthwise HP transfer, a feat thought impossible by Yang et al. [7].

- **Optimal shape and compute efficiency.** For SP, μP, and $\alpha = \{0.5, 1\}$, we construct scaling laws over models ranging from 75M to 1.9B total parameters, trained at a compute-optimal frontier of 20 tokens per parameter (TPP) [3]. We use this to revisit the question of compute-efficient transformer *shapes* (i.e. width-to-depth ratio $N : L$). The 1.9B parameter models (i.e. with 1.5B non-embedding parameters) trained with $\alpha = 1$ achieve 11.8% and 34.4% FLOP savings over μP for optimally-shaped and 179-layer models, respectively (Figure 4).

- **Refined desiderata for HP transfer.** In Section 6, we describe the differences between SP, μP, and $\alpha \in \{0.5, 1\}$. Given the strong empirical evidence for HP transfer for $\alpha = 1$, we devise a refined set of desiderata for HP transfer that includes the notion of *complete feature learning*: we require that the learned representation in every layer of the model remains non-lazy [10] (i.e. non-linear) with respect to the parameter in both that layer and all earlier ones. Only $\alpha = 1$ ensures complete feature learning, thus we name it CompleteP.

- **Extended parameterizations for modern training.** Modern LLMs train with pre-LN and AdamW. In Table 1 and Appendix D, we extend existing parameterizations [8] to include prescriptions for LayerNorm learning rates $\eta$, weight decay, bias, and AdamW $\epsilon$ as functions of depth and width. These changes are **essential** for stable training with $\alpha = 0.5$ (Figure 7).

## 2 Related work

**Theoretical approach to HP tuning.** Early methods for selecting HPs analyzed networks at initialization to ensure numerical stability in the initial forward and backward passes [11–28]. Subsequent work devised two parameterizations with consistent training dynamics at infinite width: the *Neural Tangent Kernel (NTK)* parameterization in which the model converges to its linearization around initialization [29–31] and the *mean-field/μP* parameterization [4, 6, 8, 10, 32–39]. Training at the simultaneous limits of width and depth was explored in [40, 41] for fully connected networks, in [6, 7] for vanilla ResNets and in [8] for transformers. Infinite-limit descriptions of training in the mean-field/μP parameterization are difficult to study analytically but give a clear proposal for HP transfer by requiring consistent dynamics of hidden layer representations across model scale [41–44, 32].

This approach to HP transfer was taken up in [6–8], which derive a family of mean-field parameterizations for ResNets indexed by $\alpha \in [0.5, 1]$ (see Table 1). Based on a heuristic related to feature diversity, [7] argued HP transfer was not truly possible at any $\alpha$ but that $\alpha = 0.5$ was the best in practice. In contrast, we show CompleteP ($\alpha = 1$) yields both HP transfer and FLOP savings during pre-training and provide theoretical justification in Section 6. Other interesting mean-field approaches to HP transfer include adaptations to sharpness-aware optimization [45], to low-precision training [46], to sparse training [47], and finally to state space models [48].

**Empirical approaches to depth scaling.** Empirical approaches to HP transfer in [49, 50] normalize layer outputs and learning rates to yield HP transfer across width, consistent with the μP prescription. HP transfer across depth is often more ad hoc: [49] proposes something resembling $\alpha = 0.5$, while Large et al. [50] proposes $\mathbf{h}^{\ell+1} = \frac{L-1}{L} \cdot \mathbf{I} + L^{-1} \cdot \mathcal{F}_\ell(\mathbf{h}^\ell)$ per layer, reminiscent of setting $\alpha = 1$. Several other works, though not directly focused on HP transfer, aim to stabilize training in deep ResNets through LN modifications. Sun et al. [51] advocates multiplying pre-LN output in layer $\ell$ by $\ell^{-0.5}$, similar to taking $\alpha = 0.5$. Similarly, [52] applies LN to both the input and output of each residual block, avoiding exponential-in-depth activation and gradient variance. Finally, [53] adjusts the scale of weights in each residual block, up-weights residual branch contributions, and inserts LN after the residual add. While this allows one to train very deep networks, it does not achieve depth HP transfer.

**Compute-optimal transformer $N : L$ ratios.** Theoretical and empirical work have studied the optimal transformer shape. Notably Kaplan et al. [2], using SP and large-scale tests, showed a wide range of $N : L$ were close to compute-optimality, with $N : L \approx 100$ being the optimal ratio. This finding guides modern LLM shapes, which often fix $N : L \approx 100$ when parameters and tokens are scaled [54–72]. Yet SP does not fairly admit stable width and depth scaling, which undermines the prior conclusion. McLeish et al. [73] adopt a $\eta = \eta_{\text{base}}/\sqrt{L}$ parameterization resembling incomplete $\alpha = 0.5$ and study compute-optimal $N : L$. However, their parameterization does not admit training stability (Figure 7) or HP transfer so their deeper models were disadvantaged, causing them to conclude shallower models are optimal. In Section 5 we revisit the transformer $N : L$ study with proper width and depth scaling control with CompleteP and show that even $N : L \approx 10$ remains close to compute optimality. Rather than fixing $N : L$, other works perform empirical searches to estimate scaling exponents for $N$ and $L$ [74, 75]. Levine et al. [76] propose a theory for transformer capacity based on separation rank, predict that optimal $N : L$ increases with parameter count, and provide empirical validation. Mixture of experts (MoE) [77] enable transformer capacity to scale without increasing depth. Similarly, [78] propose parallel sub-networks and argue that large depth isn't necessary for competitive performance. We focus on dense transformers and consider MoE and parallel sub-networks as out of scope.

## 3 Methodology

For all experiments in this work, we train decoder-only Transformer language models [79] with pre-normalization, untied embeddings, ALiBi position embeddings [80] and ReLU$^2$ nonlinearity [81, 82], using the AdamW optimizer with decoupled weight decay [83] with $\beta_1 = 0.9, \beta_2 = 0.95, \epsilon = 1e{-}16$. The learning rate $\eta$ schedule follows a linear warmup of min(10% of steps, 375M tokens), then a linear decay to zero [84]. We pretrain our models using an autoregressive loss (i.e. the next token prediction objective) on the SlimPajama dataset [85] with a maximal sequence length of 2048 tokens using the GPT-2 tokenizer [79]. All models use $d_{\text{head}} = 64$ for each attention head and

feedforward dimension $4N$. For all parameterizations, we use $\mathbf{Q}^\top\mathbf{K}/N$ as proposed in [5] to account for correlation between $Q$ and $K$. We use $N_{\text{base}} = 256$, $L_{\text{base}} = 2$ and scale parameters according to Table 1. All experiments were performed using Cerebras CS-3 systems. We refer the reader to Appendix G for full methodology of all experiments.

Table 1 provides an overview of how to implement the parameterizations we test in this paper.[2] We consider models with depth $L$ (or $2L$ residual blocks to account for both MLP and attention blocks), and width $N$ (e.g., residual activations $h^\ell \in \mathbb{R}^N$). We define adjustments in terms of width multiplier $m_N = N/N_{\text{base}}$ and depth multiplier $m_L = L/L_{\text{base}}$, where $N_{\text{base}}, L_{\text{base}}$ are the width and the depth of a base model. For the base model, i.e., when $m_N = 1, m_L = 1$, all parameterizations are equivalent. We use $N_{\text{base}} = 256, L_{\text{base}} = 2$. In experiments where only $m_N = 1$, SP is equivalent to μP; such results are labelled "SP/μP". We extend the infinite depth parameterizations in the literature [6–8] with corrections for bias learning rate $\eta$, LayerNorm $\eta$, AdamW $\epsilon$, and weight decay $\lambda$. See Appendix D for derivations of these extensions and practical advice for applying CompleteP to different architectures. Appendix E for empirical verification of $\epsilon$ scaling[3]. We provide a minimal implementation of Table 1 which reproduces Figure 7 at: https://github.com/EleutherAI/nanoGPT-mup/tree/completep[4].

Table 1: Summary of SP, μP, and $\alpha \in \{0.5, 1\}$ for a pre-LN transformer language model. Terms related to width and depth control are highlighted in orange and green respectively. Additional tunable parameters are highlighted in blue. *Hidden* refers to all linear layers in the transformer backbone.

| Parameterization | SP | μP | $\alpha \in \{0.5, 1\}$ |
|---|---|---|---|
| Emb. Init. Var. | $\sigma^2_{\text{base}}$ | $\sigma^2_{\text{base}}$ | $\sigma^2_{\text{base}}$ |
| Emb. LR (AdamW) | $\eta_{\text{base}}$ | $\eta_{\text{base}}$ | $\eta_{\text{base}}$ |
| Pre-LN Init. Var. | $\sigma^2_{\text{base}}$ | $\sigma^2_{\text{base}}$ | $\sigma^2_{\text{base}}$ |
| Pre-LN LR (AdamW) | $\eta_{\text{base}}$ | $\eta_{\text{base}}$ | $\eta_{\text{base}} m_L{}^{\alpha-1}$ |
| Hidden Init. Var. | $\sigma^2_{\text{base}}$ | $\sigma^2_{\text{base}} \cdot m_N^{-1}$ | $\sigma^2_{\text{base}} \cdot m_N^{-1}$ |
| Hidden LR (AdamW) | $\eta_{\text{base}}$ | $\eta_{\text{base}} \cdot m_N^{-1}$ | $\eta_{\text{base}} \cdot m_N^{-1} \cdot m_L{}^{\alpha-1}$ |
| Hidden Bias LR (AdamW) | $\eta_{\text{base}}$ | $\eta_{\text{base}}$ | $\eta_{\text{base}} m_L{}^{\alpha-1}$ |
| Hidden WD (AdamW) | $\lambda_{\text{base}}$ | $\lambda_{\text{base}} \cdot m_N$ | $\lambda_{\text{base}} \cdot m_N$ |
| MHA Residual | $\mathbf{X}^l + \text{MHA}(\text{LN}(\mathbf{X}^l))$ | $\mathbf{X}^l + \text{MHA}(\text{LN}(\mathbf{X}^l))$ | $\mathbf{X}^l + m_L{}^{-\alpha} \cdot \text{MHA}(\text{LN}(\mathbf{X}^l))$ |
| MLP Residual | $\mathbf{Z}^l + \text{MLP}(\text{LN}(\mathbf{Z}^l))$ | $\mathbf{Z}^l + \text{MLP}(\text{LN}(\mathbf{Z}^l))$ | $\mathbf{Z}^l + m_L{}^{-\alpha} \cdot \text{MLP}(\text{LN}(\mathbf{Z}^l))$ |
| Final-LN Init. Var. | $\sigma^2_{\text{base}}$ | $\sigma^2_{\text{base}}$ | $\sigma^2_{\text{base}}$ |
| Final-LN LR (AdamW) | $\eta_{\text{base}}$ | $\eta_{\text{base}}$ | $\eta_{\text{base}}$ |
| Unemb. Init. Var. | $\sigma^2_{\text{base}}$ | $\sigma^2_{\text{base}}$ | $\sigma^2_{\text{base}}$ |
| Unemb. LR (AdamW) | $\eta_{\text{base}}$ | $\eta_{\text{base}}$ | $\eta_{\text{base}}$ |
| Unemb. Fwd. | $\mathbf{X}^L \mathbf{W}_{\text{unemb}}^\top$ | $\mathbf{X}^L \mathbf{W}_{\text{unemb}}^\top \cdot m_N^{-1}$ | $\mathbf{X}^L \mathbf{W}_{\text{unemb}}^\top \cdot m_N^{-1}$ |
| AdamW $\epsilon$ (Residual blocks) | $\epsilon_{\text{base}}$ | $\epsilon_{\text{base}} \cdot m_N^{-1}$ | $\epsilon_{\text{base}} \cdot m_N^{-1} \cdot m_L{}^{-\alpha}$ |
| AdamW $\epsilon$ (Emb. & Unemb.) | $\epsilon_{\text{base}}$ | $\epsilon_{\text{base}} \cdot m_N^{-1}$ | $\epsilon_{\text{base}} \cdot m_N^{-1}$ |

## 4 Depth-wise HP transfer and $\alpha$

Here, we investigate the HP transfer abilities of μP and $\alpha \in \{0.5, 1\}$ as model depth $L$ is varied.

**Traditional HP transfer** First, we investigate the depth-wise transfer of $\eta_{\text{base}}$ and $\sigma_{\text{base}}$ while training all models for 300M tokens with batch size $B = 128$ and $\lambda_{\text{base}} = 0$. We remark that this is the traditional setting where the phenomenon of HP transfer was originally observed [5]. Figure 2 shows that SP, μP, and $\alpha = 0.5$ do not have stable optimal $\eta_{\text{base}}$ or $\sigma_{\text{base}}$ as depth $L$ is varied. When

---

[2]For the embedding layer, we choose an initialization variance scale independent of input dimension due to the input data being one-hot encoded. If the input data is dense, then we would require a pre-factor of $N_{\text{in}}^{-1/2}$.

[3]In practice, we observe that a "small enough" $\epsilon$ is sufficient to achieve consistent dynamics and transfer across width depth. See Appendix G for details.

[4]Thanks to [86] for finding typos in our Table 1 AdamW $\epsilon$ emb. & unemb. prescriptions and mistakes in our official code! We have updated Table 1 and our official code to reflect the $\epsilon_{\text{base}} \cdot m_N^{-1}$ prescriptions from Equation (40) and [87].

using the optimal HPs for $L = L_{\text{base}} = 2$, the right column shows these parameterizations do not achieve consistent loss improvements with depth.

*Finding 1: With SP, μP, and $\alpha = 0.5$, as L is varied, models do not share the same optimal HPs.*

For CompleteP ($\alpha = 1$), the optimal $\eta_{\text{base}}$ and $\sigma_{\text{base}}$ remain stable across depths as evidenced by concentric curves with stable minima. It is the only parameterization to consistently improve loss for deeper models without HP tuning, as shown in Figure 2 right column. Such stable HP transferability dramatically reduces HP tuning budgets for deep models. We demonstrate HP transfer from 2 to 128 layers, exceeding the depth of LLaMA-70B (80 layers) and LLaMA-405B (126 layers) [58].

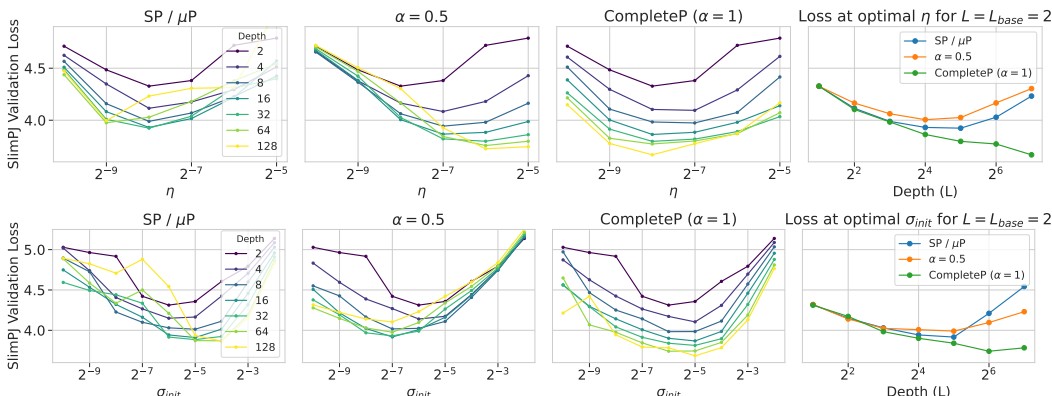

Figure 2: Depth-wise HP transfer, with 300M training tokens. **Top:** Learning rate ($\eta$) transfer. **Bottom:** Initialization standard deviation ($\sigma_{\text{init}}$) transfer. See Table 4 for experimental details.

**Compute-efficient HP transfer** The parameterizations we test are grounded in theories derived under a fixed token count [8]. We now investigate whether the depth-wise transfer of $\eta_{\text{base}}$ extends to the compute-optimal setup prescribed in Hoffmann et al. [3], where all the models are trained for 20 TPP. We also select compute-efficient batch sizes based on total training FLOPs [2, 88–90], and ensure a well-tuned $\lambda_{\text{base}}$ [91] (see Appendix G for further details). Figure 3 shows this compute-optimal setup is much less sensitive to the choice of $\eta_{\text{base}}$ compared to training for 300M tokens with $B = 128$ and $\lambda_{\text{base}} = 0$ in Figure 2. Despite this reduced sensitivity, the right column shows $\alpha = 1$ achieves superior losses to SP, μP, and $\alpha = 0.5$ without additional $\eta_{\text{base}}$ tuning from $L_{\text{base}}$. These results lead us to conclude:

*Finding 2: Only CompleteP ($\alpha = 1$) achieves reliable depth-wise HP transfer.*

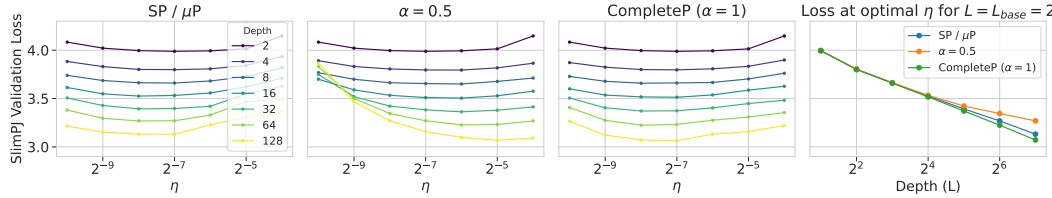

Figure 3: Learning rate transfer test under compute-optimal setting (20 TPP, batch size based on BS-FLOP power law, optimal weight decay $\lambda_{\text{base}}$). See Table 4 for experimental details.

*Finding 3: Empirically, we observe larger TPP reduces HP sensitivity (Figure 2 vs. 3).*

## 5 Re-examining compute-optimal N:L ratio

We now turn to the study of the optimal $N : L$ ratio in transformers, as its tuning can significantly impact compute efficiency. Kaplan et al. [2] popularized the practice of fixing $N : L \approx 100$ and scaling both model parameters and tokens [54–72], but without adopting a parameterization that

allows for infinite width and depth control. The consequences of this are exemplified in Section 4, where we have shown that deeper models suffer under SP/μP due to hyperparameter detuning and instability, leading $\alpha = 1$ to outperform. This depth-disadvantage represents an important confounding variable in these studies on the optimal $N:L$. In other words, without an adequate depth parameterization, deeper models are not given their best chance to succeed. In this section, we conduct the first study of compute-optimal $N:L$ with simultaneous control of infinite depth and width and show that $\alpha = 1$ enables a wider range of close-to-compute-optimal aspect ratios, where even $N:L \approx 10$ remain close to compute-optimal.

**Experimental setup** We train models in the compute-optimal setting of 20 TPP, select compute-efficient batch sizes based on FLOPs [2, 88–90], and ensure well-tuned $\eta, \lambda, \sigma$ for $L_{\text{base}} = 2$ [91] as in the previous section. See Appendix G for extensive experimental details and plotting methodology. This time, we vary $N, L$ to approximately maintain the total number of non embedding parameters $P_{\text{non-emb}} = 12N^2L$, for values of $P_{\text{non-emb}} \in \{50M, 300M, 1.5B\}$. This setup matches the model sizes from Kaplan et al. [2], but while they use the same number of tokens for all runs (compute-inefficient), we follow the compute-optimal prescription of Hoffmann et al. [3]. Instead of pure web text [2], we train on SlimPajama: a mix of web text, academic prose, and code. We use the maximal update parameterization meaning our HPs have superior tuning with respect to changing width compared to Kaplan et al. [2] who use the standard parameterization. These differences modernize our setup and make our results more applicable to contemporary LLM training.

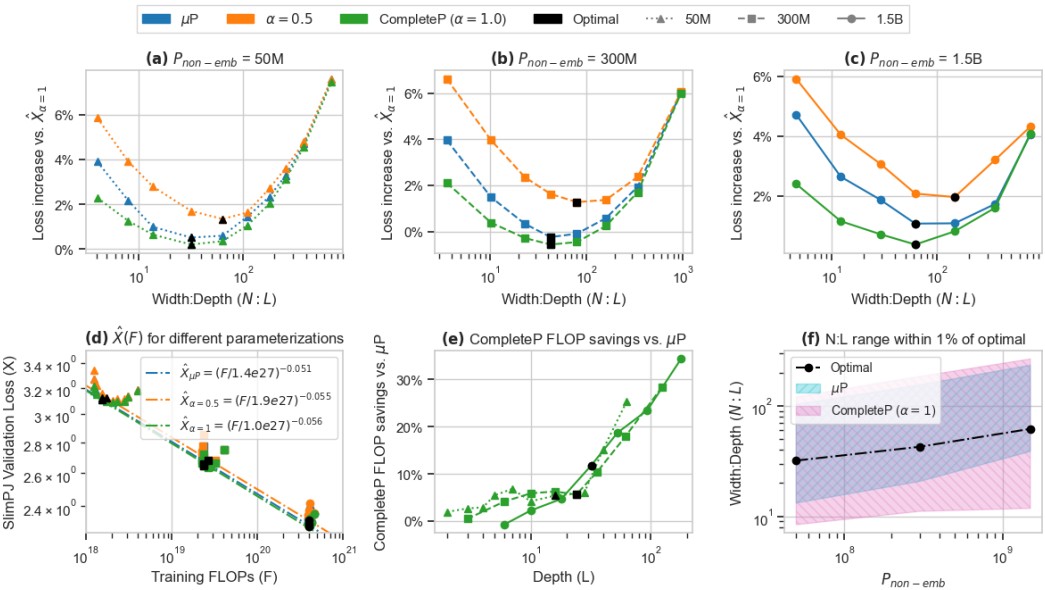

Figure 4: Optimal $N:L$ across model sizes. **(a)-(c)** For models of size $\{50M, 300M, 1.5B\}$ we see an increase in optimal $N:L$ as size increases but optimal $N:L$ is less than $\approx 60$ in this range of model sizes. **(d)** Scaling laws with FLOPs for the optimal aspect ratios in each parameterization. **(e)** FLOP savings of CompleteP compared to μP baseline, as a function of depth $L$. **(f)** Shaded regions represent the $N:L$ range with $\leq 1\%$ loss increase relative to compute-optimal $N:L$.

**Results** We first fit the power law of the form $\hat{X}(F) = (F/a)^{-b}$ describing how validation cross entropy loss ($X$) scales with a power law in FLOPs ($F$) [1, 2, 62]. The scaling law $\hat{X}(F)$, reported in Figure 4d is fitted with the most compute-efficient $N:L$ from each parameter count and it defines the compute-efficient frontier in our setup. To fairly compare model loss obtained with different FLOP budgets, we then evaluate each $N:L$ configuration in terms of the distance to the fitted $\hat{X}_{\alpha=1}(F)$. This approach is similar to Figure 5 from Kaplan et al. [2]; in our case it provides a measure of the loss difference with respect to the reference scaling law of models trained with CompleteP. We report $N:L$ as a function of this measure in Figure 4a-c. Notice that the optimal aspect ratio is the same in μP and CompleteP ($\alpha = 1$), while $\alpha = 0.5$ prefers slightly wider models. In all cases, CompleteP ($\alpha = 1$) gives lower loss values. In particular, we observe the following trend, (see also Figure 4f) :

> ***Finding 4**: Compute-optimal $N : L$ trends larger with increasing scale, for all parameterizations.*

**Advantages of CompleteP at large depth**. The gap between µP and CompleteP increases for deeper models. We quantify this gap in terms of FLOPs savings in Figure 4e, where we can see:

> ***Finding 5**: The deeper the model, the more FLOP savings CompleteP ($\alpha = 1$) has over µP.*

We attribute this gap to the fact that at higher depth µP is more detuned due to lack of HP transfer. In the 1.5B parameter models, CompleteP saves 11.8% of FLOPs for optimal $N : L$ and 34.4% FLOPs in the deepest models (lowest $N : L$). The shaded regions in Figure 1-right and 4f represent the N:L range with $\leq 1\%$ loss increase relative to compute-optimal N:L (see Appendix G.2 for more detail).

> ***Finding 6**: As model scale increases, CompleteP ($\alpha = 1$) enables deep-narrow models (small $N : L$) to remain close to compute-optimal.*

For $P_{\text{non-emb}}$=1.5B, $\alpha = 1$ enables $N{:}L = 11.8$ to remain within 1% of compute-optimal, compared to $N{:}L = 38.7$ from µP. Hardware latency worsens with increasing depth, making shallow-wide models preferable for latency-sensitive applications. However, low-memory hardware can benefit from narrow-deep models by streaming one layer at a time into memory. Prominent examples of weight streaming exist in both training [92] and inference [93, 94] settings.

**Downstream Performance**. In Table 2 we evaluate the 20 TPP $P_{\text{non-emb}}$=1.5B models at the minimum and optimal $N : L$ settings, and confirm the upstream gains also translate to gains across five downstream tasks [95–100] that collectively test for common sense reasoning, world knowledge, and reading comprehension. The details of downstream evaluation setup are in Appendix G.3.

Table 2: Zero-shot downstream evaluation accuracy for 20 TPP $P_{\text{non-emb}} \approx 1.5$B models at the optimal and minimum $N : L$ ratios. We report average task accuracy $\pm$ standard error. If the average task accuracy is within the standard error, both columns are **bolded**.

| Task | Random | Optimal ($N = 1984, L = 32$) | | | Deepest ($N = 832, L = 179$) | | |
|---|---|---|---|---|---|---|---|
| | | µP | $\alpha = 0.5$ | CompleteP ($\alpha = 1$) | µP | $\alpha = 0.5$ | CompleteP ($\alpha = 1$) |
| SlimPJ Val. xent. ($\downarrow$) | 4.701 | 2.302 | 2.326 | **2.286** | 2.386 | 2.417 | **2.330** |
| FLOP savings vs. µP ($\uparrow$) | - | 0% | -19.6% | **11.8%** | 0% | -23.7% | **34.4%** |
| HellaSwag ($\uparrow$) | 25.0 | 53.3 $\pm$ 0.5 | 51.7 $\pm$ 0.5 | **54.2** $\pm$ 0.5 | 49.1 $\pm$ 0.5 | 46.7 $\pm$ 0.5 | **52.7** $\pm$ 0.5 |
| ARC-Easy ($\uparrow$) | 25.0 | 54.4 $\pm$ 1.0 | 54.5 $\pm$ 1.0 | **55.6** $\pm$ 1.0 | 50.0 $\pm$ 1.0 | 49.2 $\pm$ 1.0 | **54.6** $\pm$ 1.0 |
| LAMBADA ($\uparrow$) | 0.0 | **54.3** $\pm$ 0.7 | 51.7 $\pm$ 0.7 | **54.9** $\pm$ 0.7 | 51.8 $\pm$ 0.7 | 43.8 $\pm$ 0.7 | **53.3** $\pm$ 0.7 |
| RACE ($\uparrow$) | 25.0 | **34.9** $\pm$ 1.5 | **34.7** $\pm$ 1.5 | **34.1** $\pm$ 1.5 | 33.5 $\pm$ 1.5 | 32.7 $\pm$ 1.5 | **35.6** $\pm$ 1.5 |
| PIQA ($\uparrow$) | 50.0 | **70.7** $\pm$ 1.1 | **71.6** $\pm$ 1.1 | **71.5** $\pm$ 1.1 | **69.6** $\pm$ 1.1 | 70.5 $\pm$ 1.1 | **70.6** $\pm$ 1.1 |
| BoolQ ($\uparrow$) | 50.0 | 58.4 $\pm$ 0.9 | **61.3** $\pm$ 0.9 | 60.7 $\pm$ 0.9 | 57.8 $\pm$ 0.9 | 57.9 $\pm$ 0.9 | **59.0** $\pm$ 0.9 |
| Downstream Avg. ($\uparrow$) | 29.2 | 54.3 $\pm$ 0.3 | 54.3 $\pm$ 0.3 | **55.2** $\pm$ 0.3 | 52.0 $\pm$ 0.3 | 50.1 $\pm$ 0.3 | **54.3** $\pm$ 0.3 |

Given the popularity of TPP>20 training for enhanced parameter efficiency [56], we also compare compare parameterizations at 200 TPP for $P_{\text{non-emb}} \in \{50M, 300M\}$ in Table 3. CompleteP ($\alpha = 1$) consistently achieves the best loss across all parameterizations, even in the 200 TPP regime.

Table 3: SlimPJ validation loss for $P_{\text{non-emb}} \in \{50M, 300M\}$ models trained for 200 TPP.

| 50M optimal ($N = 512, L = 16$) | | | 50M deepest ($N = 256, L = 63$) | | | 300M optimal ($N = 1024, L = 24$) | | | 300M deepest ($N = 448, L = 125$) | | |
|---|---|---|---|---|---|---|---|---|---|---|---|
| µP | $\alpha$=0.5 | CompleteP ($\alpha$=1) | µP | $\alpha$=0.5 | CompleteP ($\alpha$=1) | µP | $\alpha$=0.5 | CompleteP ($\alpha$=1) | µP | $\alpha$=0.5 | CompleteP ($\alpha$=1) |
| 2.867 | 2.874 | **2.859** | 2.973 | 2.998 | **2.950** | 2.451 | 2.455 | **2.443** | 2.537 | 2.534 | **2.497** |

## 6 Desiderata for Hyperparameter Transfer

Our empirical results suggest that, among all the parameterizations we tested, $\alpha = 1$ yields the best performance in terms of both HP transfer and pre-training efficiency. In this section, we provide a theoretical heuristic for why this might be and how one might have arrived *a priori* at the CompleteP ($\alpha = 1$) parameterization. Our heuristic consists of proposing three desiderata for constructing a good parameterization. While variants of our first two desiderata were already proposed in [4, 7, 8], Desideratum 3 is novel and distinguishes $\alpha = 1$. To frame our discussion, recall that parameterizations are typically constructed to ensure consistent, numerically stable, and meaningful updates for both the hidden layer representations and the network outputs, during training at any model size.

**Setup** Since we are interested in finding such a parameterization for transformers, let us consider a residual network with $L$ residual layers of width $N$ in which the representation $h^\ell \in \mathbb{R}^N$ of a fixed input after $\ell$ residual layers satisfies the following recursion

$$\mathbf{h}^{\ell+1} = \mathbf{h}^\ell + L^{-\alpha}\, \mathcal{F}_\ell(\mathbf{h}^\ell; \boldsymbol{\theta}^\ell), \qquad \ell = \{1, \ldots, L\}. \tag{2}$$

The residual block $\mathcal{F}_\ell$ is a fixed depth neural network such as an attention or MLP block. The value of $\alpha$ determines at what scale residual blocks contribute to the residual stream. The desiderata below constrain $\alpha \in [0.5, 1]$, consistent with prior work on depth transfer [6–8]. We will denote by $\theta_\ell$ the trainable parameters in layer $\ell \in [L] = \{1, \ldots, L\}$ and consider updates of the form

$$\boldsymbol{\theta}^\ell \leftarrow \boldsymbol{\theta}^\ell + \Delta\boldsymbol{\theta}^\ell, \qquad \Delta\boldsymbol{\theta}^\ell = -\eta^\ell \cdot \mathbf{g}_{\mathrm{AdamW}}^\ell. \tag{3}$$

Here $g_{\mathrm{AdamW}}$ is the AdamW update and, as we shall see from Desideratum 2, we shall have to take

$$\eta^\ell = \Theta(L^{\alpha-1})$$

to ensure that the change in $\mathbf{h}^\ell$ entries is $\Theta(1)$ for any depth or width (see Appendix C.2 or [8]). We focus our presentation on the AdamW optimizer because it is widely used for LLM training, but the desiderata below can be equally well applied to any optimizer (see [6–8]).

**Stable Initialization.** Our first desideratum is a numerical stability requirement for hidden layer representations $h^\ell$ and network outputs $f$ at initialization.

**Desideratum 1** (Stable Initialization)**.** *Hidden layers and output remain stable at initialization. More precisely, for all layers $\ell \in [L]$, $\frac{1}{N}\|\mathbf{h}^\ell\|_2^2 \in \Theta(1)$ and $f \in O(1)$, as $N \to \infty, L \to \infty$.*

This desideratum prescribes how to scale weight variances (or pre-factors) as in Table 1. It also constrains the value of $\alpha$ to be at least 0.5 (see Appendix C.1). It can be viewed as a numerical stability condition and has been well-studied in the signal-propagation literature [11, 16, 17, 19–21, 40]. We review why this desideratum imposes $\alpha \geq 1/2$ in Appendix C.1.

**Maximal Residual Stream Update.** Many parameterization schemes, including those from works on signal propagation [16, 17, 27, 12], the NTK parameterization, and the Mean Field / $\mu$P approaches [29, 33, 4], satisfy Desideratum 1 for width scaling $N \to \infty$ at fixed $L$. The core distinction between them is the presence or absence of feature learning. While there are several ways to make this precise, we follow the original $\mu$P work [4] and formalize feature learning by considering the change in hidden layer representations after each step of training.

Consider the simple case of fixed-depth fully connected networks $\mathbf{h}^{\ell+1} = \mathbf{W}^\ell \phi(\mathbf{h}^\ell)$ in which the change $\Delta\mathbf{h}^{\ell+1}$ to first order in the learning rate can be naturally written as a sum of two terms:

$$\Delta\mathbf{h}^{\ell+1} = \Delta\mathbf{W}^\ell \phi(\mathbf{h}^\ell) + \mathbf{W}^\ell \Delta\phi(\mathbf{h}^\ell).$$

The first term captures the change in $\Delta\mathbf{h}^{\ell+1}$ from the immediately preceding weights $\mathbf{W}^\ell$ and the second term reflects the change in $\mathbf{h}^{\ell+1}$ from updates to representations in previous layers.

To derive a parameterization for HP transfer across width, [4] required not only that the relative change $\|\Delta\mathbf{h}^{\ell+1}\|_2/\|\mathbf{h}^{\ell+1}\|_2$ to pre-activations in layer $\ell$ be $\Theta(1)$ but also that the proportion of this change $\|\Delta\mathbf{W}^\ell \phi(\mathbf{h}^\ell)\|_2/\|\mathbf{h}^{\ell+1}\|_2$ attributable directly to the parameters $W_\ell$ must be $\Theta(1)$ as well. This excludes degenerate situations such as when one trains only the first few layers of a network.

While the maximal update requirement leads to a unique parameterization for HP transfer across width in a fixed-depth network, it is not directly applicable to the setting where one also scales the network depth. For example, in the case when $\alpha = 0.5$ and $\mathcal{F}_\ell(\mathbf{h}^\ell; \boldsymbol{\theta}^\ell) = \mathbf{W}^\ell \phi(\mathbf{h}^\ell)$, one must actually require $\|\Delta\mathbf{W}^\ell \phi(\mathbf{h}^\ell)\|^2/\|\mathbf{h}^{\ell+1}\|_2^2 = \Theta(L^{-0.5})$ in order for activations to remain stable in the sense that $\|\Delta\mathbf{h}^{\ell+1}\|/\|\mathbf{h}^{\ell+1}\| = \Theta(1)$ (see Appendix C.2 & [6, 7]). To cover residual networks of growing depth, we therefore require that the maximal update prescription hold *per residual block*.

**Desideratum 2** (Maximal Residual Stream Update)**.** *Each residual block's weights should contribute order $1/L$ to feature movements, and each non-residual block should contribute constant order. More precisely, for all $\ell \in [L-1]$, each block's parameter update $\boldsymbol{\theta}^\ell \mapsto \boldsymbol{\theta}^\ell + \Delta\boldsymbol{\theta}^\ell$ should contribute the change $\frac{1}{N}\|\Delta_{\boldsymbol{\theta}^\ell}\mathbf{h}^{\ell+1}\|_2^2 \in \Theta(1/L)$. Moreover, for the embedding and unembedding layers we require $\frac{1}{N}\|\Delta\mathbf{W}^0 x\|_2^2 \in \Theta(1)$ and $\frac{1}{N}\|\Delta\mathbf{W}^L \mathbf{h}^L\|_2^2 \in \Theta(1)$.*

Desideratum 2 matches the maximal update prescription of [4] when depth $L$ is finite, as $\Theta(1/L) = \Theta(1)$. Furthermore, Desideratum 2 uniquely determines the depth-dependence of the learning rate in AdamW to be $\eta = \Theta(L^{1-\alpha})$ for the update to satisfy the $\Theta(1/L)$ scale [6]. Note also that requiring $\mathcal{F}(h^\ell; \theta_\ell) = \Theta(1)$ with respect to $L$ constrains $\alpha \leq 1$. When combined with Desideratum 1 this explains why we consider $0.5 \leq \alpha \leq 1$. We also emphasize that additional care must be taken to correctly determine learning rates for biases and LayerNorm (LN) parameters, otherwise $\alpha = 0.5$ fails Desideratum 2 for a pre-LN transformer (see Figure 7 and Appendix D).

**Complete Feature Learning.** Variants of Desiderata 1 and 2 have been proposed in prior work and are satisfied by any $\alpha \in [0.5, 1]$. In this section we provide one possible intuition for what distinguishes $\alpha = 1$ and why it might work so well in our experiments. We do so by showing that only $\alpha = 1$ gives parameterization in which every layer remains uniformly non-linear in its parameters, regardless of depth and width (see Desideratum 3). More precisely, we define the **linearization** $h^{\text{lin},\theta}$ of a function $\mathbf{h}(\theta)$ with respect to $\theta$ about $\theta_0$ is

$$\mathbf{h}^{\text{lin},\theta}(\boldsymbol{\theta}, \boldsymbol{\theta}_0) = \mathbf{h}(\boldsymbol{\theta}_0) + \langle \nabla_{\boldsymbol{\theta}} \mathbf{h}(\boldsymbol{\theta})|_{\boldsymbol{\theta}_0}, \boldsymbol{\theta} - \boldsymbol{\theta}_0 \rangle.$$

We say $\mathbf{h}$ is **linear** in $\boldsymbol{\theta}$ if, $\mathbf{h} = \mathbf{h}^{\text{lin},\theta}$.

**Definition.** *We say a layer $\mathbf{h}^\ell$ is **lazy** with respect to a subset of parameters $\theta \subset \{\theta_j\}_{j<\ell}$ if at finite depth and width, $\mathbf{h}^\ell$ is not linear in $\theta$ and the change $\Delta_\theta \mathbf{h}^\ell$ at initialization from updating only $\theta$ (i.e. replacing $\theta \mapsto \theta + \Delta\theta$) is asymptotically the same as the change to the linearization of $h$:*

$$\frac{|\Delta_\theta \mathbf{h}^\ell - \Delta_\theta \mathbf{h}^{lin,\theta}_\ell|}{|\Delta_\theta \mathbf{h}^{lin,\theta}_\ell|} = o(1), \quad \text{as } N, L \to \infty. \tag{4}$$

**Desideratum 3** (Complete Feature Learning). *The network parameterization satisfies complete feature learning, i.e. neither the hidden layers $\{h^\ell\}_{\ell \in [L]}$ nor the model output $f$ are lazy with respect to any subset of model parameters.*

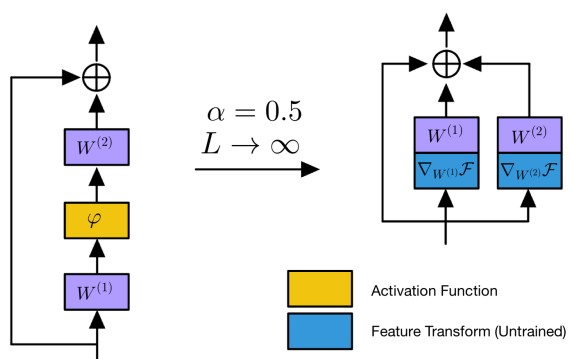

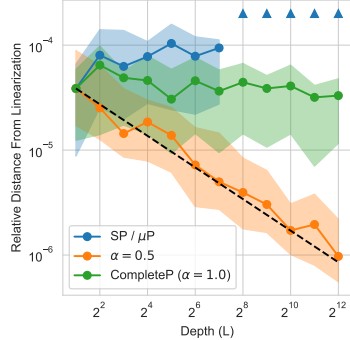

Figure 5: As $L \to \infty$, $\alpha = 0.5$ asymptotically linearizes residual blocks (Eqn. 2). **Left:** Depth 2 MLP residual block. For $\alpha = 0.5$, this block is only nonlinear with respect to parameters when $L$ is finite. **Right:** Linearized MLP residual block, with $\nabla_{W^{(k)}}\mathcal{F}$ a non-trainable transform of $h^\ell$.

Figure 6: Only $\alpha = 1$ achieves complete feature learning with stability. SP/$\mu$P diverges at large depth (triangles are NaNs), while $\alpha = 0.5$ converges to the linearization at a rate of $1/\sqrt{L}$. The y axis is the left hand side of Eq. 4.

To put this definition into context, note that in the NTK regime, where we do not scale depth, the model output are lazy with respect to all non-output parameters $\{\theta_\ell\}_{\ell=0}^{L-1}$ [101]. As we will illustrate through a simple example, only $\alpha = 1$ gives complete feature learning for the $L \to \infty$ limit.

**Simple Block Depth 2 Example** To illustrate why Desideratum 3 distinguishes between $\alpha = 1$ and other values of $\alpha$, consider a width $N = 1$ toy model where we only scale depth $L \to \infty$

$$h^{\ell+1} = h^\ell + L^{-\alpha} W^\ell_{(2)} W^\ell_{(1)} h^\ell, \qquad \ell = \{1, \ldots, L\} \tag{5}$$

To satisfy Desiderata 1 and 2, we need the weight update to satisfy $\Delta w^i_\ell = \Theta(L^{\alpha-1})$ for both $i = 1, 2$. Next, consider the update to $h^\ell$ from the parameters of preceding residual block parameters:

$(W_{(1)}^\ell, W_{(2)}^\ell) = \boldsymbol{\theta}^\ell \mapsto \boldsymbol{\theta}^\ell + \Delta\boldsymbol{\theta}^\ell$. Taylor expanding yields:

$$\Delta_{\boldsymbol{\theta}^\ell} h^{\ell+1} = \langle \nabla_{\boldsymbol{\theta}^\ell} h^{\ell+1}, \Delta\boldsymbol{\theta}^\ell \rangle + \frac{1}{2} \nabla_{\boldsymbol{\theta}^\ell}^2 h^{\ell+1}[\Delta\boldsymbol{\theta}^\ell, \Delta\boldsymbol{\theta}^\ell]$$

$$= L^{-\alpha}(W_{(2)}^\ell h^\ell \underbrace{\Delta W_{(1)}^\ell}_{L^{\alpha-1}} + W_{(1)}^\ell h^\ell \underbrace{\Delta W_{(2)}^\ell}_{L^{\alpha-1}}) + L^{-\alpha} h^\ell \underbrace{\Delta W_{(1)}^\ell \Delta W_{(2)}^\ell}_{L^{2(\alpha-1)}} \cdot \qquad (6)$$

$$\underbrace{\phantom{= L^{-\alpha}(W_{(2)}^\ell h^\ell \Delta W_{(1)}^\ell + W_{(1)}^\ell h^\ell \Delta W_{(2)}^\ell)}}_{L^{-1}} \qquad \underbrace{\phantom{L^{-\alpha} h^\ell \Delta W_{(1)}^\ell \Delta W_{(2)}^\ell}}_{L^{\alpha-2}}$$

Note the first term is exactly the change $\Delta h_{\ell+1}^{\text{lin},\theta_\ell}$ to the linearization of $h^{\ell+1}$, which has the correct order $\Theta(L^{-1})$. The second term therefore captures the difference between updating $\mathbf{h}^{\ell+1}$ and $\mathbf{h}_{\ell+1}^{\text{lin},\theta_\ell}$ and has order $\Theta(L^{\alpha-2})$. The two terms have the same order only when $\alpha = 1$. For all $\alpha < 1$, as we increase the depth $L$ of the network, the contribution of $\mathbf{h}^{\ell+1}$ to the non-linear term diminishes with scale. We empirically verify this, for the same toy model, in Figure 6 (see Appendix G.4 for details). Since we expect the optimal HPs in shallow models to depend on linear and non-linear dynamics of $\mathbf{h}_{\ell+1}$, we do not expect these HPs to be optimal when scaling depth with $\alpha < 1$.

While we carried out our computations above only in the simple toy model (Equation (5)), the form of the calculation extends to any arbitrary $\mathcal{F}_\ell(\mathbf{h}^\ell; \boldsymbol{\theta}^\ell)$ with bounded depth, including large wide MLPs and self-attention blocks. The core observation is that higher order terms in the Taylor expansion $(k > 1)$ of $\Delta_{\boldsymbol{\theta}^\ell} \mathbf{h}^{\ell+1}$ will always have the form

$$\nabla^k \mathbf{h}^{\ell+1} \cdot [[\Delta\boldsymbol{\theta}^\ell]^{\otimes k}] = \Theta\left(L^{-\alpha} \cdot L^{k(\alpha-1)}\right), \qquad [\Delta\boldsymbol{\theta}^\ell]^{\otimes k} = \Theta(L^{k(\alpha-1)}) \qquad (7)$$

We summarize the above discussion on lazy learning into the following statement.

> **Finding 7:** *Only CompleteP ($\alpha = 1$) ensures stable training, maximal updates, and complete feature learning as $N \to \infty, L \to \infty$.*

Given that $\alpha = 1$ is the unique $\alpha$ that ensures complete feature learning while scaling both width and depth, we dub it *CompleteP*.

## 7 Limitations

Our theoretical analysis considers scaling width and depth at fixed token count, which limits the direct applicability to the fixed TPP compute-optimal regime. However, the scaling predictions derived in the fixed token setting (Figure 2) have predicted success in the fixed TPP compute-optimal regime (Figure 3). The $\hat{X}$ fits in Section 5 were only fit to 3 points due to budget constraints with training models larger than 1.5B parameters. However the three points span a 256x FLOP range and are each the most compute-efficient point from each group of 7-10 N:L values for each parameter count (Table 5), so they are likely to be an accurate picture of the compute-efficient frontier. We use the scaling law fits for interpolation rather than extrapolation, where fitting to limited data can make predictions unreliable.

## 8 Conclusion

We studied in large pre-LN transformers a variety of parameterizations, i.e., prescriptions for adjusting hyperparameters such as learning rates as functions of network architecture. Among them, we found that CompleteP ($\alpha = 1$) gives HP transfer when varying depth and width. We showed that CompleteP achieves significant FLOP savings during pre-training compared to other parameterizations, even in compute-optimal settings with jointly scaled batch, dataset, and model sizes. To achieve this, we extended the theoretical analysis of CompleteP to include prescriptions for how to scale LN parameters and AdamW's $\epsilon$ across depth and width.

To understand the salutary effects of CompleteP, we formalized a set of desiderata for designing parameterizations. They include variants of desiderata from prior work as well as a novel property, which we called *complete feature learning*, that distinguishes CompleteP. This parameterization is simple to implement, and we released a public code base to facilitate reproduction and further study. Our empirical tests were conducted on models with up to 1.5B non-embedding parameters. In future work, we plan to train large, state-of-the-art LLMs that leverage CompleteP's utility in HP transfer and compute-efficient training.

## Acknowledgments and Disclosure of Funding

BB and LN thank the Google PhD fellowship for support. BH is supported by a Sloan Fellowship, NSF DMS-2143754, and NSF DMS-2133806. BB and C.P. acknowledge support by NSF grant DMS-2134157, NSF CAREER Award IIS-2239780, DARPA grant DIAL-FP-038, a Sloan Research Fellowship, and The William F. Milton Fund from Harvard University. C.P. and B.B's work has been made possible in part by a gift from the Chan Zuckerberg Initiative Foundation to establish the Kempner Institute for the Study of Natural and Artificial Intelligence.

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

## A  Broader impacts

As LLM compute budgets grow, compute efficient training methods are becoming increasingly important for reducing carbon emissions [102] and offsetting environmental and financial costs of large model training [103]. This work presents methods which improve the compute efficiency of LLM training, especially for large deep models. There is also growing recognition that HP tuning is a key contributor to these costs. HP tuning is costly, possibly undermining equity in AI research due to financial resources [104]. During model retraining, sensitivity to HPs also leads to downstream costs [104]. This work presents methods which can reduce these costs and sensitivities and thus improve equity.

## B  Coordinate check test for verification of stable training

To verify our theoretical expectations for training stability match empirical results, we perform "coordinate check" tests where we scale $L$ and measure the change in activation size at the final residual addition $h_L$ in Figure 7. Interestingly, $\alpha = 0.5$ as described in the literature [6–8] did not achieve transformer training stability as we initially expected (2nd column). We reasoned this was due to a lack of consideration for how the bias and LayerNorm parameters can affect training stability. After adopting the bias and LayerNorm $\eta$ adjustments proposed in Table 1 and Appendix D, we achieved training stability with $\alpha = 0.5$ (3rd column). Figure 7 shows both $\alpha \in \{0.5, 1\}$ achieve stability for any depth. This test is a necessary but not sufficient condition for satisfying Desiderata 2.

## C  Scalings that Guarantee Our Desiderata

In the following analysis, we argue why $\alpha \geq \frac{1}{2}$ is necessary for Desideratum 1 (Stable Initialization) and we argue for AdamW LR to be $\eta = L^{\alpha-1}$ for each residual block to achieve Desideratum 2 (Maximal Residual Stream Update).

### C.1  Desideratum 1: Stable Initialization

In this section, we demonstrate that $\alpha \geq \frac{1}{2}$ is necessary for stable signal propagation at initialization. For concreteness in the simplest model that captures this effect, we consider a residual MLP network

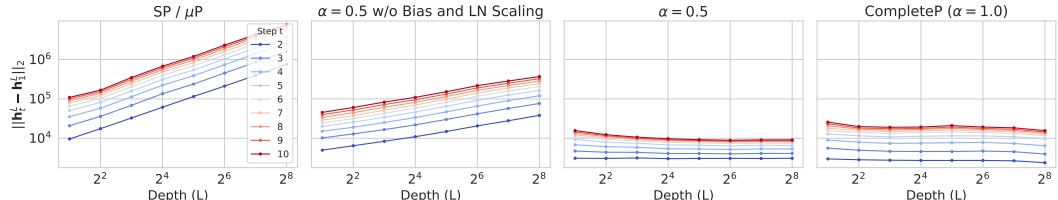

Figure 7: Coordinate check. Frobenius norm of activations after merged residual streams from MHSA and MLP blocks across models of increasing depths after training step $t$. 1st column: SP/$\mu$P without depth control. 2nd column: $\alpha = 0.5$ as described in literature [6–8]. 3rd column: $\alpha = 0.5$ (Table 1). 4th column: $\alpha = 1.0$. $\alpha \in \{0.5, 1\}$ achieve stability for any depth.

with block depth 1, the model analyzed in [6],

$$\mathbf{h}^{\ell+1} = \mathbf{h}^\ell + \frac{1}{L^\alpha}\mathbf{W}^\ell\phi(\mathbf{h}^\ell). \tag{8}$$

The weights are initialized with $W_{ij}^\ell \sim \mathcal{N}(0, \frac{\sigma_W^2}{N})$. Defining $H^\ell \equiv \frac{1}{N}|\mathbf{h}^\ell|^2$ as the norm of the residual stream variables, we have the following recursion for $H^\ell$ at large $N$

$$H^{\ell+1} = H^\ell + L^{-2\alpha}\sigma_W^2\mathbb{E}_{h\sim\mathcal{N}(0,H^\ell)}\phi(h)^2, \tag{9}$$

where the average $\mathbb{E}_h\,[\,]$ represents an average over the hidden neurons at residual layer $\ell$ [6]. Taking for concreteness $\phi(h) = \mathrm{ReLU}(h)$, the above equation gives

$$H^{\ell+1} = H^\ell + \frac{\sigma_W^2}{2}L^{-2\alpha}H^\ell = \left[1 + \frac{\sigma_W^2}{2}L^{-2\alpha}\right]^\ell H^0 \tag{10}$$

where $H^0$ is the scale of the first residual variable. The last layer variance therefore has the following large depth limit

$$\lim_{L\to\infty} H^L = \begin{cases} H^0 & \alpha > \frac{1}{2} \\ \exp\left(\frac{\sigma_W^2}{2}\right)H^0 & \alpha = \frac{1}{2} \\ \infty & \alpha < \frac{1}{2} \end{cases}. \tag{11}$$

We see that $\alpha \geq \frac{1}{2}$ is necessary for the residual stream variance to converge to a finite value as $L \to \infty$. For $\alpha = \frac{1}{2}$ the residual stream covariance depends on the activation function and the initialization variance $\sigma_W^2$ while for the $\alpha > \frac{1}{2}$, the signal propagation becomes trivial in the infinite depth limit as the contribution from the nonlinearity and the initial weight variance disappears at large $L$.

## C.2 Desideratum 2: Maximal Update

In this section, we consider the Adam learning rate necessary to achieve $\Theta(1)$ changes to the residual stream variables. First, we note that the backward pass variables $\mathbf{g}^\ell \equiv \frac{\partial\mathcal{L}}{\partial\mathbf{h}^\ell}$ are strongly correlated across layers since, from the chain rule,

$$\mathbf{g}^\ell = \mathbf{g}^{\ell+1} + \frac{1}{L^\alpha}\dot\phi(\mathbf{h}^\ell) \odot \left[\left(\mathbf{W}^\ell\right)^\top\mathbf{g}^{\ell+1}\right]. \tag{12}$$

We therefore note that any weighted average of over layers is $\Theta_L(1)$, such as

$$\frac{1}{L}\sum_{\ell=1}^L C^\ell h_i^\ell = \Theta_L(1)\,, \quad \frac{1}{L}\sum_{\ell=1}^L C^\ell g_i^\ell = \Theta_L(1), \tag{13}$$

where $C^\ell$ is an arbitrary sequence with increments of scale $C^{\ell+1} - C^\ell = \Theta_L(L^{-1})$. Consider now the update under a sign-GD update

$$W_{ij}^\ell(t+1) = W_{ij}^\ell(t) + \eta^\ell\frac{1}{Z_{ij}^\ell}g_i^{\ell+1}\phi(h_j^\ell)\,, \quad Z_{ij}^\ell = \sqrt{(g_i^{\ell+1})^2\phi(h_j^\ell)^2} \tag{14}$$

where $\eta^\ell$ is the learning rate for this hidden layer. Note that AdamW will contain the same scaling (with $N, L$) for the numerator and denominator but will also contain momentum (which does not change the scaling with $N, L$) and an additional $\epsilon$ parameter which we discuss how to scale with $N, L$ in Appendix D.5.

We now discuss how to choose $\eta^\ell$ to obtain $\Theta_{N,L}(1)$ updates to the residual stream (Desideratum 2). First, we can express the residual variables at layer $\ell + 1$ at step $t$ of training as

$$\mathbf{h}^{\ell+1}(t) = \mathbf{h}^\ell(t) + L^{-\alpha} \mathbf{W}^\ell(0) \phi(\mathbf{h}^\ell(t))$$
$$+ L^{-\alpha} \sum_{t' < t} \left[ \frac{\eta^\ell}{\mathbf{Z}^\ell(t')} \odot \mathbf{g}^{\ell+1}(t') \phi(\mathbf{h}^\ell(t'))^\top \right] \phi(\mathbf{h}^\ell(t)) \tag{15}$$

The entries of the above vector from the update to $\mathbf{W}^\ell(t)$ have the following scaling with $N$ and $L$

$$v_i^\ell(t, t') \equiv \frac{1}{N} \sum_{j=1}^N \frac{g_i^{\ell+1}(t') \phi(h_j^\ell(t'))}{Z_{ij}^\ell(t')} \phi(h_j^\ell(t)) = \Theta_{N,L}(1), \tag{16}$$

as the variables $\phi(h_j^\ell(t'))$ and $\phi(h_j^\ell(t))$ are random variables with $\Theta(1)$ correlation. We desire the final layer of the residual stream to achieve a $\Theta(1)$ updates

$$\mathbf{h}^L(t) = \mathbf{h}^0(t) + \underbrace{L^{-\alpha} \sum_{\ell=0}^{L-1} \mathbf{W}^\ell(0) \phi(\mathbf{h}^\ell(t))}_{\text{Stable by D1 for } \alpha \geq 1/2} + \underbrace{L^{-\alpha} N \sum_{\ell=0}^{L-1} \eta^\ell \sum_{t' < t} \mathbf{v}^\ell(t, t')}_{\text{Desired scale} = \Theta(1)} \tag{17}$$

Since $\alpha \geq \frac{1}{2}$, the stability of the middle term is guaranteed from the D1 analysis. The $\mathbf{v}^\ell$ variables are strongly correlated across layers, so a sum over all $L$ layers will give a result of order $\Theta(L)$. To make this final term $\Theta_{N,L}(1)$ we must choose a learning rate of scale

$$\eta^\ell = \eta_{\text{base}} N^{-1} L^{\alpha-1}. \tag{18}$$

This choice causes the final term above in the $\mathbf{h}^L(t)$ formula to be $\Theta(1)$, resulting in a $\Theta(1)$ change to the residual variables due to the weight updates as desired.

## D  Derivation for bias, LN, WD, and AdamW $\epsilon$ parameterization adjustments

### D.1  On Generalizability of the Parameterization

In the later parts of this subsection, we will discuss derivations of parameterizations for several different components of the architecture via a heuristic method, which we find very accurately represents the scaling important quantities. However, this remains limited as there are other architecture components that are not contained by our manuscript, which may require further adjustments and derivations.

With this in mind, we can divide up the future generalizations into three categories:

1. No modifications required. This includes LayerNorm at any position (pre, post, QK-norm etc.), adding biases at different locations, LR schedule, and other well-normalized Adam-like algorithms (e.g. SignGD, Adagrad, Lion), or position embeddings (learned, RoPE, ALiBi, NoPE, etc.). The reason is that these do not change the scale of both the forward pass and hidden layer updates due to training, with respect to width and depth. LayerNorm, for example, maintains the $\Theta(1)$ scale of each neuron, and as long as the update maintains the $\Theta(1)$ scale, then no changes are required.

2. Slight modifications are required and known. A good example of this is SGD, where the learning rate scaling is derived in Table 1 of Bordelon et al. [8] (up to an equivalent ABC-reparameterization). Notice here that compared to Adam, the learning rate of SGD needs to be scaled up due to the prefactors in front of both the weight matrices and the residual branch, where as these extra factors are canceled out in Adam due to normalization.

3. More theoretical derivations required. This includes MoE, long context, batch size (and schedule), gradient clipping, LAMB, and momentum. Some of these are a relatively straight forward exercise to derive (e.g. LAMB and other optimization algorithms), others are less straight forward. In particular, any scaling that requires increasing the number of data points and training steps along with width and depth remains theoretically challenging. Our approach in this work computes the theoretical scaling for finitely many data points and training steps, and testing whether or not hyperparameter transfer empirically, which has been yielding very good results (see e.g. Figure 3).

For the third category, we would effectively need to derive the scale of the updates and the downstream effects for each new architectural modification. A good example is consider the effect of bias weights in the next subsection. Here biases $\mathbf{b}^\ell$ are updated such that it contributes the same amount of changes to $\mathbf{h}^{\ell+1}$ as the usual weights $\mathbf{W}^\ell$. In order to do that, we need to choose the appropriate initialization and learning scale to satisfy this requirement. All other modifications follow the same mechanism, and the rest of this section serve as an example derivation for all future generalizations.

## D.2   Bias Parameters

In this section, we consider the effect of a bias parameter. To start, we will illustrate the effect of bias parameters in a depth 1 MLP residual block

$$f = \frac{1}{N}\mathbf{W}^L\phi(\mathbf{h}^L) \ , \ \mathbf{h}^{\ell+1} = \mathbf{h}^\ell + \frac{1}{L^\alpha}\left[\mathbf{W}^\ell\phi(\mathbf{h}^\ell) + \mathbf{b}^\ell\right] \ , \ \mathbf{h}^1 = \mathbf{W}^0\mathbf{x} \tag{19}$$

The entries of the bias vectors $\mathbf{b}^\ell$ are initialized with unit variance $\mathbf{b}^\ell \sim \mathcal{N}(0,\mathbf{I})$ for all layers while $\mathbf{W}^\ell$ entries are $\mathcal{N}(0,\frac{1}{N})$ for intermediate layers $\ell \in \{1,...,L-1\}$ and $\mathcal{N}(0,1)$ for encoder and decoder layers $\ell \in \{0,L\}$. With this choice, stable signal propagation is guaranteed for $\alpha \geq \frac{1}{2}$. To obtain the desired $\frac{1}{L}$ increment in the residual stream, the weight and bias parameters must be updated with scale

$$\Delta W_{ij}^\ell = \Theta\left(N^{-1}L^{\alpha-1}\right) \ , \ \Delta b_i^\ell = \Theta(L^{\alpha-1}) \ , \ \ell \in \{1,...,L-1\} \tag{20}$$

For the Adam optimizer this directly sets the desired scaling of learning rate for each of these parameters

$$\eta_{W^\ell} = \Theta\left(N^{-1}L^{\alpha-1}\right) \ , \ \eta_{b^\ell} = \Theta(L^{\alpha-1}). \tag{21}$$

This argument extends to more complicated blocks such as self attention layers and deeper MLPs. However, for the first and last layer of the transformer which we want to achieve $\Theta(1)$ changes to their block outputs we need to set

$$\Delta W_{ij}^\ell = \Theta\left(1\right) \ , \ \Delta b_i^\ell = \Theta(1) \ , \ \ell \in \{0,L\}. \tag{22}$$

## D.3   LayerNorm

While a static LayerNorm operation was examined in [8], most LayerNorm operations include trainable gain and bias parameters for each neuron.

$$\text{LN}^\ell(\mathbf{h}^\ell) = \frac{1}{\sqrt{\sigma_h^2 + \epsilon}}\left[\mathbf{h}^\ell - \mu_h\mathbf{1}\right] \odot \mathbf{g}^\ell + \mathbf{b}^\ell \tag{23}$$

where $\mu_h$ and $\sigma_h^2$ are the mean and variance of the entries of $\mathbf{h}^\ell$. For any hidden block in the residual stream, the entries of $\mathbf{g}$ and $\mathbf{b}$ must be updated with

$$\forall \ell \in \{1,...,L-1\} \ , \ \Delta g_i = \Theta(L^{\alpha-1}) \ , \ \Delta b_i^\ell = \Theta(L^{\alpha-1}). \tag{24}$$

while for the first and last layer (before and after the residual stream),

$$\Delta g_i = \Theta(L^{\alpha-1}) \ , \ \Delta b_i^\ell = \Theta(L^{\alpha-1}) \ , \ \ell \in \{0,L\}. \tag{25}$$

## D.4 Weight Decay

In this section, we will perform a heuristic calculation the weight decay scaling required in AdamW, which we will approximate with a SignGD-like update of the type

$$\mathbf{W}^\ell(t+1) = \mathbf{W}^\ell(t) - \eta \frac{1}{\mathbf{Z}^\ell} \odot \nabla_{\mathbf{W}^\ell} L(\theta(t)) - \eta\lambda\mathbf{W}^\ell(t) \,, \tag{26}$$

where $\mathbf{Z}^\ell$ is a collection of constant such that $\frac{1}{Z}\nabla_{\mathbf{W}}L(\theta(t))$ have $\Theta(1)$ entries with respect to width $N$ and depth $L$, $\lambda > 0$ is the weight decay parameter.

Suppose in either an MLP or inside a residual block, we have a fully connected layer (including the cases for $Q, K, V$ blocks)

$$\mathbf{h}^{\ell+1} = \mathbf{W}^\ell\phi(\mathbf{h}^\ell) \,, \tag{27}$$

where $\mathbf{W}^\ell \in \mathbb{R}^{N\times N}$ is initialized with $\mathcal{N}(0, \sigma_{\text{base}}^2/N)$.

Suppose we want $\mathbf{h}^{\ell+1}$ to change by order $\Theta(1)$ in an SignGD update with one data point, then we get that

$$\mathbf{W}^\ell(t+1) = \mathbf{W}^\ell(t) - \frac{\eta}{\mathbf{Z}^\ell} \odot \mathbf{g}^{\ell+1}(t)\phi(\mathbf{h}^\ell(t))^\top - \eta\lambda\mathbf{W}^\ell(t) \,, \tag{28}$$

where $\mathbf{Z} \in \mathbb{R}^{N\times N}$ is once again the normalization constants, and $\mathbf{g}^{\ell+1}$ is the backward pass variable [32, Equation 16]. However, in this case we have that $g$ and $\phi(h)$ are already $\Theta(1)$ in the previous iterate, so $\mathbf{Z}$ is itself $\Theta(1)$.

Next we will calculate $\Delta\mathbf{h}^{\ell+1}(t+1) = \mathbf{h}^{\ell+1}(t+1) - \mathbf{h}^{\ell+1}(t)$ to get

$$\Delta\mathbf{h}^{\ell+1} = \Delta\mathbf{W}^\ell\phi(\mathbf{h}^\ell) + \mathbf{W}^\ell\Delta\phi(\mathbf{h}^\ell) \,, \tag{29}$$

where we abuse notations slightly and drop the $t$ indices when clear. Here we will also assume as induction hypothesis that $\Delta\phi(\mathbf{h}^\ell)$ is well behaved so we will focus on the first term only, which is

$$\begin{aligned} \Delta\mathbf{W}^\ell\phi(\mathbf{h}^\ell) &= -\eta \left[ \frac{1}{\mathbf{Z}^\ell} \odot \mathbf{g}^{\ell+1}\phi(\mathbf{h}^\ell)^\top \right] \phi(\mathbf{h}^\ell) - \eta\lambda\mathbf{W}^\ell\phi(\mathbf{h}^\ell) \\ &= -\eta \left[ \frac{1}{\mathbf{Z}^\ell} \odot \mathbf{g}^{\ell+1}\phi(\mathbf{h}^\ell)^\top \right] \phi(\mathbf{h}^\ell) - \eta\lambda \underbrace{\mathbf{h}^{\ell+1}}_{\Theta(1)} . \end{aligned} \tag{30}$$

where the entries of $\mathbf{h}^{\ell+1}$ are $\Theta_{N,L}(1)$ by the arguments of Appendix C.1. Following the arguments of Appendix C.2, the learning rate $\eta$ must be set as

$$\eta = \eta_{\text{base}} \frac{L^{\alpha-1}}{N} \,, \tag{31}$$

to ensure the correct change to the residual stream variables. Since we desire the final term to be on the same scale this means that $\lambda = \Theta_{N,L}(N)$. Our proposed in weight decay scaling rule (Table 1) is also used in a few recent works [91, 105, 106]. The contribution of this section is to show no additional depth correction to the weight decay parameter $\lambda$ is required.

## D.5 Adam $\epsilon$ Parameter

In principle, the $\epsilon$ parameter for Adam may need to be scaled with width or depth to obtain stable behavior. The key problem can be seen from the update for a parameter $\theta$

$$\Delta\theta^\ell = \frac{\eta^\ell}{\sqrt{v_t^\ell} + \epsilon^\ell} m_t^\ell \tag{32}$$

where $v_t$ is a moving average of squared gradients and $m_t$ is a moving average of gradients. Prior work outlined various strategies to achieve this stability as width is scaled [87, 38]. We will now describe strategies to achieve of the $\epsilon$ parameter across depths.

*Layer-wise $\epsilon$, $\eta$:* One strategy is to utilize the current parameterization provided in the main text and to modify the $\epsilon$ parameter and learning rate for each type of layer. The parameterization is

$$f = \frac{1}{N}\mathbf{W}^L\phi(\mathbf{h}^L) \,, \quad \mathbf{h}^{\ell+1} = \mathbf{h}^\ell + \frac{1}{L^\alpha}\mathbf{W}^\ell\phi(\mathbf{h}^\ell) \,, \quad \mathbf{h}^1 = \mathbf{W}^0\mathbf{x}, \tag{33}$$

where the hidden weights $W_{ij}^\ell$ are initialized with random entries from

$$W_{ij}^\ell(0) \sim \mathcal{N}(0, N^{-1}) \,,\, \ell \in \{1, ..., L-1\} \tag{34}$$

and the first and last layer are initialized with unit variance

$$W_{ij}^0(0) \sim \mathcal{N}(0, 1) \,,\, W_{ij}^L(0) \sim \mathcal{N}(0, 1) \tag{35}$$

We note that the $m_t$ and $v_t$ variables have the following scalings in the residual stream

$$m_t^\ell = \Theta\left(L^{-\alpha} N^{-1}\right) \,,\, \sqrt{v_t^\ell} = \Theta\left(L^{-\alpha} N^{-1}\right) \,,\, \ell \in \{1, ..., L-1\} \tag{36}$$

$$m_t^\ell = \Theta\left(N^{-1}\right) \,,\, \sqrt{v_t^\ell} = \Theta\left(N^{-1}\right) \,,\, \ell \in \{0, L\} \tag{37}$$

This parameterization demands the following learning rate for hidden layers for an $\epsilon \to 0$ limit to be stable

$$\eta^\ell = \eta_0 \, N^{-1} L^{\alpha-1} \,,\, \ell \in \{1, ..., L-1\} \tag{38}$$

$$\eta^\ell = \eta_0 \,,\, \ell \in \{0, L\} \tag{39}$$

Now, to consider the effect of $\epsilon$, we desire it to match the scale of the raw gradients $m_t$ so we take

$$\epsilon^\ell = \epsilon_0 L^{-\alpha} N^{-1}.$$
$$\epsilon^\ell = \epsilon_0 N^{-1} \,,\, \ell \in \{0, L\}. \tag{40}$$

This will ensure that $\epsilon$ is of the same order as $\sqrt{v_t}$ for all hidden layers. For the read-in and readout we can pre-multipy gradients by a constant.

*Reparameterize the Defintion of the Model:* Another strategy that one could adopt is to reparameterize the model so that a single value of $\epsilon$ can be used for every layer. In this case,

$$f = \frac{1}{NL^\alpha} \mathbf{W}^L \phi(\mathbf{h}^L) \,,\, \mathbf{h}^{\ell+1} = \mathbf{h}^\ell + \frac{1}{L^\alpha} \mathbf{W}^\ell \phi(\mathbf{h}^\ell) \,,\, \mathbf{h}^1 = \frac{1}{L^\alpha} \mathbf{W}^0 \mathbf{x}, \tag{41}$$

where the readin and readout weights are initialized as

$$W_{ij}^0(0) \sim \mathcal{N}(0, L^{2\alpha}) \,,\, W_{ij}^L(0) \sim \mathcal{N}(0, L^{2\alpha}). \tag{42}$$

The global $\epsilon$ parameter can now be set as

$$\epsilon = \epsilon_0 L^{-\alpha} N^{-1}. \tag{43}$$

The learning rates must be set as

$$\eta^\ell = \eta_0 L^\alpha \,,\, \ell \in \{0, L\} \tag{44}$$

$$\eta^\ell = \eta_0 L^{\alpha-1} N^{-1} \,,\, \ell \in \{1, ..., L-1\}. \tag{45}$$

These rules will ensure the correct scale updates in each layer.

## E   Depth-wise transfer of AdamW $\epsilon$

Everett et al. [38] show appropriate AdamW $\epsilon$ scaling is important for compute-efficient large models. Building on the width adjustment for AdamW $\epsilon$ introduced by Yang and Littwin [87], we add a depth correction for AdamW $\epsilon$. Figure 8 shows this correction enables stable training for a wider range of $\epsilon_{\text{base}}$.

## F   Depth Scaling Training Dynamics

We illustrate the training dynamics in the form of loss curves in Figure 9 and 10, which slice the $\eta$ transfer test in Figure 2 at CompleteP ($\alpha = 1$)'s and $\alpha = 0.5$'s optimal $\eta$ respectively. CompleteP ($\alpha = 1$) consistently achieves lower training loss when model depth scales up. In $\alpha = 0.5$, we point out the loss "crossing" behavior where deeper models continually take longer steps to scale better than shallower models. This aligns with our observation in Figure 2 where $\alpha = 0.5$ desires larger $\eta$ with increasing model depth and laziness (defined in Section 6).

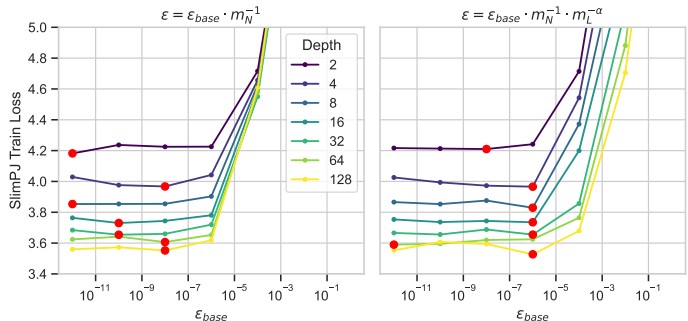

Figure 8: Using ComplteP ($\alpha = 1$), we ablate the effect of the depth-wise Adam $\epsilon$ scaling rule. The scaling rule enables a wider range of $\epsilon_{\text{base}}$ to achieve competitive loss.

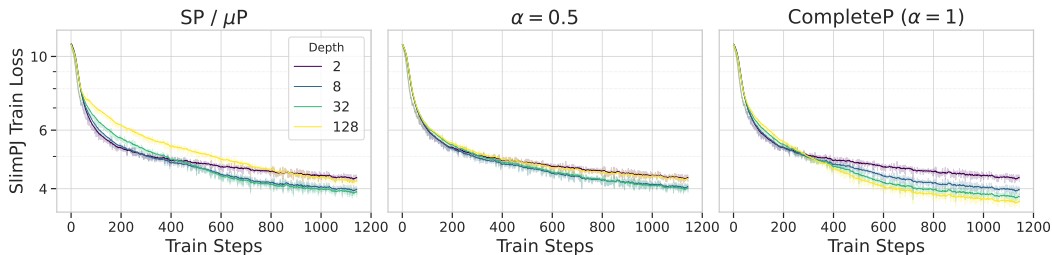

Figure 9: Training dynamics of Figure 2 for all parameterizations at CompleteP ($\alpha = 1$)'s optimal $\eta = 2^{-8}$.

## G  Additional experimental details

Table 4 contains extensive details for all experiments. The rest of this section describes the remaining experimental details.

### G.1  Compute-efficient setup

In Figure 4, to study the compute-optimal setting, first our training setup must represent compute-optimal. Following [3], we train models for 20 tokens per parameter (TPP) to ensure a compute-optimal tradeoff of parameters and tokens. We also scale our batch size according to a power law in FLOPS. Since, McCandlish et al. [88], Kaplan et al. [2] showed the critical batch size scales with a power law in loss, and loss scales with a power law in FLOPS, the critical batch size scales with a power law in FLOPS [89, 90]. We follow Equation 46 based on empirical fits and rounded to the nearest multiple of 8, where $B, F$ are batch size and training FLOPS respectively.

$$B = \max(32, 0.7857 \cdot F^{0.1527} - 306.8) \tag{46}$$

We adopt the $\sigma_{\text{base}} = 0.02$ and the optimal $\eta_{\text{base}}$ for $L = L_{\text{base}} = 2$ from Figure 2: $\eta_{\text{base}} = 0.0039$.

To ensure optimal weight decay $\lambda$ as we scale model and dataset size, we follow Wang and Aitchison [91] by setting $\lambda_{\text{base}}$ to maintain the optimal $\tau_{\text{EMA}} = (\eta_{\text{base}}\lambda_{\text{base}}n_{\text{steps}})^{-1}$. A major shortcoming of $\tau_{\text{EMA}}$ is that it does not take the learning rate schedule into account, even though learning rates are decayed in practice. Despite this we observe reliable transfer of $\tau_{\text{EMA}}$ across model sizes, provided we use the same learning rate schedule. For SP and µP, Figure 11 validates that the claims of Wang and Aitchison [91] that the optimal $\tau_{\text{EMA}}$ remains stable across different learning rates and weight decay values. We then tune $\tau_{\text{EMA}}$ for SP, µP, $\alpha = 0.5$, and CompleteP ($\alpha = 1$). Figure 12 shows $\tau_{\text{EMA}} = 0.1407$ close to optimal for all parameterizations and this adopted throughout this paper. When a non-zero weight decay is adopted, we use $\tau_{\text{ema}}, \eta_{\text{base}}, n_{\text{steps}}$ to decide the $\lambda_{\text{base}}$ used in each run.

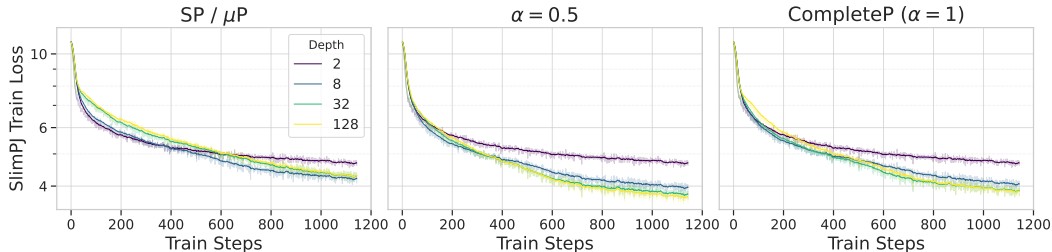

Figure 10: Training dynamics of Figure 2 for all parameterizations at $\eta = 2^{-6}$, where $\alpha = 0.5$ first demonstrates deeper-model-lower-loss ordering.

Table 4: Additional experimental details. Cells marked as "vary" mean the quantity is varied.

| Figure | $N$ | $L$ | $\sigma_{\text{init}}$ | $\eta_{\text{base}}$ | $\lambda_{\text{base}}$ | $T_{\text{EMA}}$ | $B$ | Steps | Tokens | TPP |
|---|---|---|---|---|---|---|---|---|---|---|
| 1 left, 2 top, 9, 10 | 256 | Vary | 0.02 | Vary | 0 | N/A | 128 | 1144 | 300M | Vary |
| 2 bottom | 256 | Vary | Vary | 2E-08 | 0 | N/A | 128 | 1144 | 300M | Vary |
| 3 | 256 | Vary | 0.02 | Vary | Vary | 0.1407 | Eqn. 46 | Vary | Vary | 20 |
| 4, 13, 1 middle & right | Vary | Vary | 0.02 | 2E-08 | Vary | 0.1407 | Eqn. 46 | Vary | Vary | 20 |
| 7 | 256 | Vary | 0.06 | 2E-03 | 0 | N/A | 4 | 10 | 82K | Vary |
| 8 | 256 | Vary | 0.02 | 2E-08 | 0 | N/A | 128 | 1144 | 300M | Vary |
| 11 | Vary | 6 | 0.09 | Vary | Vary | Vary | Eqn. 46 | Vary | Vary | 20 |
| 12 | Vary | Vary | 0.02 | 2E-08 | Vary | Vary | Eqn. 46 | Vary | Vary | 20 |
| Table 3 | Vary | Vary | 0.02 | 2E-08 | Vary | 0.1407 | Eqn. 46 | Vary | Vary | 200 |

## G.2 N:L study details

Here we include additional experimental details for Figure 4. We vary $N, L$ to approximately maintain $P_{\text{non-emb}} = 12N^2L$. Table 5 contains all model shapes, token counts, and FLOPs used in Figure 4. Approximations for training FLOPs like $F = 6ND$ do not account for embedding, attention, LN, nonlinearity or bias FLOPs. We use a very granular function to measure FLOPs. Although we maintain $P_{\text{non-emb}}$, the FLOPs are not equal mainly because of embedding and attention costs. We train $P_{\text{non-emb}} \in \{50M, 300M, 1.5B\}$ models, matching sizes from [2], but all at 20 TPP. [2] used a similar study design, but with the same number of tokens for all runs, meaning it didn't follow the compute-optimal prescription of [3].

For each $(F, X)$ point in Figure 4d, we calculate the "Loss increase vs. $\hat{X}_{\alpha=1}$" as $d = \frac{X - \hat{X}_{\alpha=1}}{X}$ and plot it in Figure 4a-c. To understand how the FLOP savings in Figure 4e are calculated, first recall the functional form of $\hat{X}(F) = (F/a)^{-b}$. Inverting yields $F(X) = aX^{-\frac{1}{b}}$. Due to the scale invariance of power laws, we can calculate:

$$\text{FLOP savings vs. } \mu\text{P} = 1 - \left(1 - (d_{\mu P} - d_{\alpha=1})\right)^{-1/b} \tag{47}$$

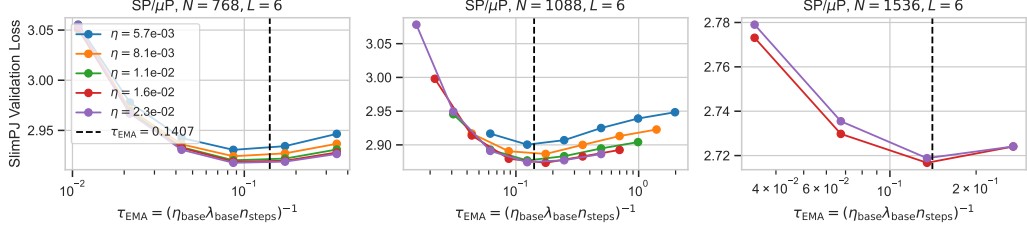

Figure 11: When grid searching learning rate $\eta_{\text{base}}$ and weight decay $\lambda_{\text{base}}$, the optimal $\tau_{\text{EMA}}$ remains stable.

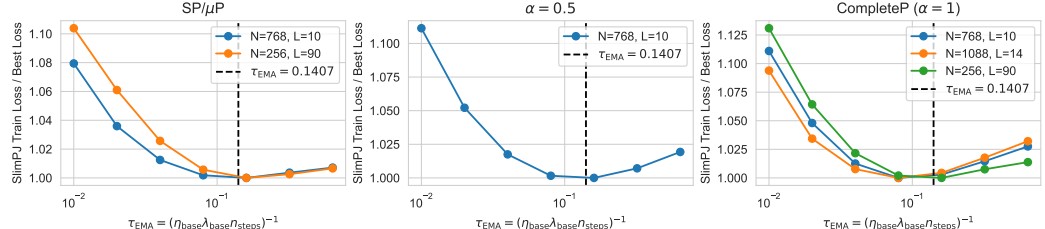

Figure 12: $\tau_{\text{EMA}}$ sweep for µP and $\alpha \in \{0.5, 1\}$. $\tau_{\text{EMA}}$ changes minimally across parameterizations. The $\tau_{\text{EMA}}$ values we empirically chose sit in the bowl of optimal loss values. $\tau_{\text{EMA}}$ for $\alpha = 0.5$ seems good but for µP and $\alpha = 1$ it seems slightly off. Looks like aspect ratio affects the optimal $\tau_{\text{EMA}}$.

Table 5: Details of model sizes used in the aspect ratio study. The compute-optimal configurations for µP and $\alpha = 1$ are **bolded**. The compute-optimal configurations for $\alpha = 0.5$ are *italicized*

| Label | $P_{\text{non-emb}}$ (M) | $P_{\text{total}}$ (M) | Tokens (B) | Train FLOPS | Steps | B | N | L | N:L |
|---|---|---|---|---|---|---|---|---|---|
| 50M | 49.8 | 75.5 | 1.5 | 1.25E+18 | 4849 | 152 | 256 | 63 | 4.1 |
| 50M | 49.3 | 81.5 | 1.6 | 1.26E+18 | 5235 | 152 | 320 | 40 | 8.0 |
| 50M | 49.7 | 88.3 | 1.8 | 1.34E+18 | 5671 | 152 | 384 | 28 | 13.7 |
| **50M** | **50.4** | **101.9** | **2.0** | **1.56E+18** | **5923** | **168** | **512** | **16** | **32.0** |
| *50M* | *49.2* | *113.6* | *2.3* | *1.76E+18* | *6301* | *176* | *640* | *10* | *64.0* |
| 50M | 49.6 | 126.8 | 2.5 | 2.07E+18 | 6730 | 184 | 768 | 7 | 109.7 |
| 50M | 48.2 | 138.3 | 2.8 | 2.35E+18 | 7033 | 192 | 896 | 5 | 179.2 |
| 50M | 50.4 | 153.3 | 3.1 | 2.82E+18 | 7198 | 208 | 1024 | 4 | 256.0 |
| 50M | 47.8 | 163.6 | 3.3 | 3.11E+18 | 7397 | 216 | 1152 | 3 | 384.0 |
| 50M | 47.6 | 189.1 | 3.8 | 4.02E+18 | 7696 | 240 | 1408 | 2 | 704.0 |
| 300M | 301.8 | 346.8 | 6.9 | 2.38E+19 | 8301 | 408 | 448 | 125 | 3.6 |
| 300M | 305.3 | 369.6 | 7.4 | 2.32E+19 | 8846 | 408 | 640 | 62 | 10.3 |
| 300M | 299.4 | 383.1 | 7.7 | 2.27E+19 | 9352 | 400 | 832 | 36 | 23.1 |
| **300M** | **302.3** | **405.2** | **8.1** | **2.38E+19** | **9699** | **408** | **1024** | **24** | **42.7** |
| *300M* | *314.8* | *443.5* | *8.9* | *2.70E+19* | *10214* | *424* | *1280* | *16* | *80.0* |
| 300M | 307.4 | 468.2 | 9.4 | 2.85E+19 | 10784 | 424 | 1600 | 10 | 160.0 |
| 300M | 302.1 | 508.0 | 10.2 | 3.20E+19 | 11275 | 440 | 2048 | 6 | 341.3 |
| 300M | 298.7 | 588.2 | 11.8 | 4.06E+19 | 12169 | 472 | 2880 | 3 | 960.0 |
| 1.5B | 1488.8 | 1572.5 | 31.4 | 4.10E+20 | 19195 | 800 | 832 | 179 | 4.6 |
| 1.5B | 1498.4 | 1614.2 | 32.3 | 3.96E+20 | 19903 | 792 | 1152 | 94 | 12.3 |
| 1.5B | 1501.6 | 1656.0 | 33.1 | 3.91E+20 | 20418 | 792 | 1536 | 53 | 29.0 |
| **1.5B** | **1512.3** | **1711.8** | **34.2** | **3.99E+20** | **21106** | **792** | **1984** | **32** | **62.0** |
| *1.5B* | *1487.9* | *1751.6* | *35.0* | *4.00E+20* | *21597* | *792* | *2624* | *18* | *145.8* |
| 1.5B | 1487.3 | 1841.1 | 36.8 | 4.26E+20 | 22474 | 800 | 3520 | 10 | 352.0 |
| 1.5B | 1487.0 | 1943.8 | 38.9 | 4.62E+20 | 23262 | 816 | 4544 | 6 | 757.3 |

Figure 13 shows how Figure 4f was created by fitting cubic functions to data $\{x = \log(N : L), y = d - d_{\alpha=1}^{\text{optimal}}\}$ and finding intersections with $y = 1$. Cubic functions were used instead of quadratics to enable better fits to functions without an axis of symmetry.

### G.3   Downstream evaluation details

We evaluated our models on downstream tasks using Eleuther Harness Evaluation (EEH) [107]. The evaluation details are reported in Table 6.

### G.4   Figure 6 details

To visualize the notion of laziness in (4):

$$\frac{\Delta_\theta \mathbf{h}^\ell - \Delta_\theta [\mathbf{h}^\ell]^{\text{lin},\theta}}{\Delta_\theta [\mathbf{h}^\ell]^{\text{lin},\theta}} = o(1), \quad \text{as } N, L \to \infty, \tag{48}$$

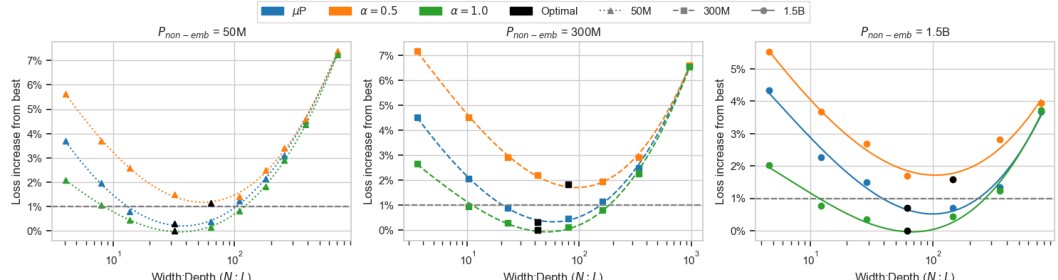

Figure 13: Plotted loss increase from best point. Fit cubic functions to log-transformed data. Found intersections with $y = 1$ for Figure 4f.

Table 6: Details on downstream tasks evaluations. **acc** stands for accuracy. **acc_norm** stands for accuracy normalized by target string's byte-length.

| Evaluation | EEH Task Name | Metric |
|---|---|---|
| Hellaswag | hellaswag | acc_norm |
| ARC-easy | arc_easy | acc_norm |
| LAMBADA | lambada_openai | acc |
| RACE | race | acc |
| PIQA | piqa | acc_norm |
| BoolQ | boolq | acc |

we consider the toy architecture of a residual network with two layers per residual branch:

$$\mathbf{h}^{\ell+1} = \mathbf{h}^\ell + L^{-\alpha}\, \mathbf{W}^\ell_{(2)}\mathbf{W}^\ell_{(1)}\mathbf{h}^\ell, \qquad \ell = \{1, \ldots, L\}, \tag{49}$$

with width $N = 256$, and perform the learning rate updates according to our parameterizations, i.e.:

$$\eta = \begin{cases} \eta_0 L^{\alpha-1} & \alpha \in [1/2, 1] \\ \eta_0 & \mu\text{P}. \end{cases} \tag{50}$$

We then perform a single gradient step only with respect to $\mathbf{W}^\ell_{(2)}$ for a fixed layer $\ell$, and calculate the left hand side of the laziness equation above for varying depths across 50 seeds per run. We report the median with lines and shade the region between the first and third quartiles. We compute $\Delta_\theta \mathbf{h}^{lin,\theta}_\ell$ explicitly by differentiating the block with respect to $\mathbf{W}^\ell_{(2)}$ and evaluate at the initial parameter values, i.e. we compute:

$$\mathbf{h}^{\text{lin},\theta}(\boldsymbol{\theta}, \boldsymbol{\theta}_0) = \mathbf{h}(\boldsymbol{\theta}_0) + \langle \nabla_\theta \mathbf{h}(\theta)|_{\boldsymbol{\theta}_0}, \boldsymbol{\theta} - \boldsymbol{\theta}_0 \rangle,$$

where $\boldsymbol{\theta} = \mathbf{W}^\ell_{(2)}$. We use $\eta_0 = 0.0001$ and train on MNIST [108]. We provide the code to reproduce this plot in supplementary material.

