# OpenReview forum: "Don't be lazy: CompleteP enables compute-efficient deep transformers"
_NeurIPS.cc/2025/Conference — NeurIPS 2025 poster_

### Official Review · Reviewer_8H9A · 2025-06-20

**Clarity:** 2
**Significance:** 3
**Originality:** 3
**Rating:** 5
**Confidence:** 3

**Summary:**

The paper studies compute-efficient hyper-parameter transfer for LLM models. A novel parameterization method called CompleteP is introduced, providing findings and guidance for efficient and reliable HP transfer. This method addresses the challenge of transferring optimal hyperparameters across varying model depths in deep transformers, which is crucial for efficient training and deployment of LLMs. Three major contributions include non-lazy depth-wise HP transfer, significant FLOPs saving when comparing with previous SOTA methods, and larger range of model width/depth ratios to remain compute-efficiency, making it suitable for various hardware settings and model deployment.

**Questions:**

1. In Figure 1, the FLOP saving is calculated based on equation (48). Are there any other hardware related metrics, such as power/ energy saving or latency profiling can be reported to show the effectiveness the compute efficiency?
2. For learning rate transfer and initialization standard deviation transfer, are there more plots beyond the task of SlimPJ? Is the trend consistent across different validation tasks?
3. Could you please provide more details on the Methodology section, for example, which model are you using, and what are the steps to implement the parameterization?
4. For Table 2, do you have results for the case of N:L > 100?

**Ethical Concerns:**

["NO or VERY MINOR ethics concerns only"]

**Final Justification:**

This paper is well studied on the topic of efficient HP transfer, with in-depth analysis, extensive experiments and empirical results. All my concerns are well addressed, so I am pleased to raise the score to 5. One additional suggestion is to include a comprehensive analysis of the generalizability of the CompleteP scaling to enhance its practical applicability.

**Limitations:**

Yes, the authors have adequately addressed the limitations.

**Paper Formatting Concerns:**

Please start a new page for Appendix.

**Quality:**

3

**Strengths And Weaknesses:**

Strengths:

This paper is well studied on the topic of efficient HP transfer, with in-depth analysis, extensive experiments and empirical results, providing practical guidance for LLM model training under different hardware requirements. The paper is well written.


Weaknesses:

1. Although the paper has in-depth study on the topic, the result is limited to pre-LN transformers. Whether the findings still hold for other architectures or optimizers remains unknown. And how the method scale to models beyond 1.5B needs to be further verified. This may limit the application of such method.
2. Some results and plots are based on empirical fitting instead of real profiling on certain hardware. This may also hurt the accuracy and generality of the report.

---

> ### Author Rebuttal · Authors · 2025-07-31
>
> Thank you very much for the helpful comments, and for your kind words regarding the paper's extensive empirical results, practical guidance, and clear presentation.
>
> ```
> The result is limited to pre-LN transformers. Whether the findings still hold for other architectures or optimizers remains unknown.
> ```
> This is a really good question! We can definitely do a better job at explaining the generalizability of the CompleteP scaling for practitioners, as well as explain the math for how to derive other cases. There are roughly three separate categories of generalizability:
>
> 1. No modifications required. This includes LayerNorm at any position (pre, post, QK-norm etc.), adding biases at different locations, LR schedule, and other well-normalized Adam-like algorithms (e.g. SignGD, Adagrad, Lion), or position embeddings (learned, RoPE, ALiBi, NoPE, etc.). The reason is that these do not change the scale of both the forward pass and hidden layer updates due to training, with respect to width and depth. LayerNorm, for example, maintains the $\Theta(1)$ scale of each neuron, and as long as the update maintains the $\Theta(1)$ scale, then no changes are required.
>
> 2. Slight modifications are required and known. A good example of this is SGD, where the learning rate scaling is derived in Table 1 of Bordelon et al. [8] (up to an equivalent ABC-reparameterization). Notice here that compared to Adam, the learning rate of SGD needs to be scaled up due to the prefactors in front of both the weight matrices and the residual branch, where as these extra factors are canceled out in Adam due to normalization.
>
> 3. More theoretical derivations required. This includes MoE, long context, batch size (and schedule), gradient clipping, LAMB, and momentum. Some of these are a relatively straight forward exercise to derive (e.g. LAMB and other optimization algorithms), others are less straight forward. In particular, any scaling that requires increasing the number of data points and training steps along with width and depth remains theoretically challenging. Our approach in this work computes the theoretical scaling for finitely many data points and training steps, and testing whether or not hyperparameter transfer empirically, which has been yielding very good results (see e.g. Figure 3).
>
> To improve the paper, we will include a more comprehensive and pedagogical section on generalizability, so can improve the accessibility of adopting CompleteP in a variety of training scenarios. Thank you for raising these concerns!
>
> ```
> How the method scale to models beyond 1.5B needs to be further verified. This may limit the application of such method.
> ```
> Thank you for noting our extensive empirical evaluation. While we agree that testing at even larger scales (10B+ parameters) would be valuable, we emphasize that our current experiments already support the following key claims:
>
> 1. CompleteP enables depth-wise HP transfer: We demonstrate HP transfer from 2 to 128 layers in Figures 2 and 3, exceeding the depth of LLaMA-70B (80 layers) and LLaMA-405B (126 layers), with additional theoretical grounding in the infinite-depth limit [8].
>
> 2. CompleteP ensures complete feature learning: We provide both theoretical justification (Section 6) and empirical results extending to 4096-layer models (Figure 6).
>
> 3. CompleteP improves the compute-efficiency of narrow-deep models: Kaplan et al. [2] performed a highly influential study that shaped the default $N:L \approx 100$ design choice across nearly all subsequent LLMs, despite relying on standard parameterization, fixed-token training, and models up to only 1.5B parameters (line 91-93). In contrast to Kaplan et al. [2], we adopt a significantly more robust setup: $\mu$P and CompleteP parameterizations, compute-efficient training (20 tokens per parameter) (Hoffmann et al. [3]) (lines 158-160). Our largest models reach 1.8B parameters, a scale that is already representative of widely-used small open models (e.g., Gemma-2B, Phi-1.5, and LLaMA-3.2-1B). With experiments spanning a 370x compute range (1.25e18-4.62e20 FLOPs, Table 5), CompleteP consistently improves the compute efficiency of deeper models.
>
> We recognize many of these points could be made more clear in the paper and will revise the results and limitations sections to better highlight this context and clarify how our results substantiate these claims. Thank you for helping to improve our submission!
>
> ```
> Some results and plots are based on empirical fitting instead of real profiling on certain hardware. This may also hurt the accuracy and generality of the report. In Figure 1, the FLOP saving is calculated based on equation (48). Are there any other hardware related metrics, such as power/ energy saving or latency profiling can be reported to show the effectiveness of the compute efficiency?
> ```
> Yes, we study FLOPs because it is hardware-agnostic and widely-used (e.g. Kaplan et al. [2], Hoffmann et al. [3]). However, we agree it is valuable to discuss other metrics and how they may inform potential hardware-specific limitations to running different N:L ratios. For example if a particular combination of code base and hardware cluster struggled to get consistent utilization across different N:L ratios, then a more hardware-specific training cost metric such as wall clock time on your cluster would provide a more accurate picture.
>
> ```
> For learning rate transfer and initialization standard deviation transfer, are there more plots beyond the task of SlimPJ? Is the trend consistent across different validation tasks?
> ```
> We agree that it is important to carefully quantify the impact of datasets on the performance of different parameterizations. Unfortunately we only evaluated our learning rate transfer and initialization standard deviation transfer tests using SlimPajama validation loss as the y-axis. However, in Table 2 we sho the SlimPajama validation loss improvements of CompleteP ($\alpha=1$) over $\alpha=0.5$ and $\mu$P carry over to six downstream tasks: HellaSwag, ARC-Easy, LAMBADA, RACE, PIQA, and BoolQ. These results suggest the CompleteP benefits should generalize beyond the just the SlimPajama dataset.
>
> ```
> Could you please provide more details on the Methodology section, for example, which model are you using, and what are the steps to implement the parameterization?
> ```
> Thank you for raising this point. The Methodology section does briefly explain we train decoder-only transformer language models with pre-normalization but then refers the reader to Appendix I which contains an extensive description of the models and training settings we used. In retrospect this seems to abrupt and we agree the Methodology section would benefit from repeating some of the Appendix I content. Regarding the steps to implement the parameterization, we do provide Table 1 in the Methodology section (line 108-109) which contains detailed steps to implement the parameterization and refer to an attached minimal code implementation on line 118. We will update line 118 to mention the code implements Table 1 and reproduces Figure 7. Please let us know if we could make the implementation steps more clear.
>
> ```
> For Table 2, do you have results for the case of N:L > 100?
> ```
> Thank you for the question. Unfortunately we don't have results for the case of N:L > 100 in Table 2 currently. However, as shown in Figures 4a, 4b, 4c, and 4e, models with higher N:L ratios (i.e., wider and shallower networks) tend to have CompleteP performance that closely tracks that of $\mu$P, as the depth $L$ approaches the base depth $L_{\text{base}}$. While we did not include specific results for N\:L > 100 in Table 2, we focused on ranges that highlight two regimes of interest: the optimal N:L ratio from a compute-efficiency standpoint, and the low N:L regime, where CompleteP uniquely provides benefits not seen with $\mu$P. We agree that further exploration of very high N:L values could be interesting and appreciate the suggestion.

---

> > ### Comment · Reviewer_8H9A · 2025-08-01
> > **Response to rebuttal**
> >
> > I would like to thank the authors for their detailed response. My concerns have been properly addressed, and I have no further question.
> >
> > Additionally, I suggest including a comprehensive analysis of the generalizability of the CompleteP scaling to enhance its practical applicability.
> >
> > Thanks.

---

### Official Review · Reviewer_sp8Y · 2025-07-01

**Clarity:** 3
**Significance:** 2
**Originality:** 2
**Rating:** 4
**Confidence:** 3

**Summary:**

The work builds on the maximal update parameterization (μP) framework by studying different values of the depth scaling parameter α in the residual connection formula: h^(ℓ+1) = h^ℓ + L^(-α) F^ℓ(h^ℓ).

The authors compare different parameterizations: Standard Parameterization (SP), μP, α = 0.5, and CompleteP (α = 1). Through extensive experiments on transformer language models ranging from 75M to 1.8B parameters trained on SlimPajama, they demonstrate that only CompleteP achieves reliable depth-wise hyperparameter transfer while maintaining training stability. Key empirical findings include: (1) Provides 11.8-34.4% FLOP savings over μP for deep models, and (2) Better accuracy (validation loss and zero shot results) at optimal points. (3) Wider range of depth to width ratios, enabling better choices across different HW platforms.

The authors also provide a theoretical justification of why setting α=1 works.

**Questions:**

1. Modern day LLMs are moving towards longer context, MoEs, Group Query Attention, RMSNorm and other enhancements to architecture. Would these analysis persists for these architectures too?

2. Can the authors provide guidance on hyperparameter sensitivity analysis for practitioners for completeP?

3. Will the benefits persist under different training regimes (e.g., longer training, different batch size schedules)?

**Ethical Concerns:**

["NO or VERY MINOR ethics concerns only"]

**Final Justification:**

After reading the rebuttal and review from other reviewers, I have increased my rating by 1. Some of the questions around novelty and scale still remain, thus I did not push my rating further.

**Quality:**

3

**Strengths And Weaknesses:**

Strengths:
1. Comprehensive empirical evaluation: The paper provides extensive experiments across multiple model sizes, training settings, and evaluation metrics, including both upstream pretraining loss and downstream task performance. Studies also cover iso-tokens and compute optimal training (20 tokens per parameter) settings, accounting for different variations

2. Theoretical grounding: The three desiderata provide a principled framework for understanding why CompleteP works, with the "complete feature learning" concept offering novel theoretical insight.


Weakness:

1. Adding α on top of μP to enable depth scaling has been covered by prior work. While this work, does consolidate discrepancies between various work and adopt it to transformers, but overall approach builds incrementally on existing μP and depth scaling work.
        --- NOTE: I did not evaluate the theoretical justification provided in Section 6. The theoretical analysis falls beyond my expertise. So any novelty or originality there would have been missed by me.

2. Largest scale of the model is 1.8B. Lack of validation on truly large-scale models (10B+ parameters) weakens the claims a bit

3. While compute optimal settings provide good insights, modern day small LLMs are trained far beyond the 20TPP number. Impact of data can be seen from Figure 2 bottom to Figure 3. This makes the claim a little harder to verify.

---

> ### Author Rebuttal · Authors · 2025-07-31
>
> Thank you for your very useful suggestions, your positive comments regarding the comprehensive empirical evaluation and theoretical insights.
>
> ```
> Adding $\alpha$ on top of $\mu P$ to enable depth scaling has been covered by prior work. While this work, does consolidate discrepancies between various work and adopt it to transformers, but overall approach builds incrementally on existing $\mu P$ and depth scaling work. --- NOTE: I did not evaluate the theoretical justification provided in Section 6. The theoretical analysis falls beyond my expertise. So any novelty or originality there would have been missed by me.
> ```
> You are completely right that various choices of $\alpha$ ranging from $1/2$ to $1$ were proposed in prior work [6-8] as ways to stabilize ResNet training across growing depth. There are, however, three sources of significant novelty related to the choice of parameterization in our work. Your comment helped us understand that we need to do a better job of highlighting them in our revision. We summarize them here:
> - **We showed depth-wise HP transfer for Transformers is possible at all.** Prior work -- notably [7] -- suggested that *no setting of $\alpha$* can possibly give HP transfer across growing depth for Transformers. In contrast we believe that we've conclusively shown that $\alpha = 1$ does in fact give good HP transfer and efficient pre-training when scaling both model depth and width.
> - **We refined prior $\alpha$ parameterizations.** We found that to ensure stable training for infinite-depth models, we needed to *refine prior parameterizations for depth scaling* to include adjustments for the bias and LayerNorm learning rates (see Section C and Table 1). We also obtained new prescriptions for scaling the AdamW $\epsilon$ parameter, which had been previously overlooked as a parameter that needs to be rescaled as a function of depth and width (see Table 1 and Figure 8).
> - **We propose a criterion for constructing good parameterizations.** In Section 6, our desideratum 3 regarding *complete feature learning* is a novel conceptual idea. It gives the intuitive explanation for why $\alpha = 1$ is most likely to give HP transfer in practice and achieve superior compute-efficiency to lazy parameterizations like $\alpha=0.5$.
>
> ```
> Largest scale of the model is 1.8B. Lack of validation on truly large-scale models (10B+ parameters) weakens the claims a bit
> ```
> Thank you for noting our comprehensive empirical evaluation. While we agree that testing at even larger scales (10B+ parameters) would be valuable, we emphasize that our current experiments already support the following key claims:
>
> 1. CompleteP enables depth-wise HP transfer: We demonstrate HP transfer from 2 to 128 layers in Figures 2 and 3, exceeding the depth of LLaMA-70B (80 layers) and LLaMA-405B (126 layers), with additional theoretical grounding in the infinite-depth limit [8].
>
> 2. CompleteP ensures complete feature learning: We provide both theoretical justification (Section 6) and empirical results extending to 4096-layer models (Figure 6).
>
> 3. CompleteP improves the compute-efficiency of narrow-deep models: Kaplan et al. [2] performed a highly influential study that shaped the default $N:L \approx 100$ design choice across nearly all subsequent LLMs, despite relying on standard parameterization, fixed-token training, and models up to only 1.5B parameters (line 91-93). In contrast to Kaplan et al. [2], we adopt a significantly more robust setup: $\mu$P and CompleteP parameterizations, compute-efficient training (20 tokens per parameter) (Hoffmann et al. [3]) (lines 158-160). Our largest models reach 1.8B parameters, a scale that is already representative of widely-used small open models (e.g., Gemma-2B, Phi-1.5, and LLaMA-3.2-1B). With experiments spanning a 370x compute range (1.25e18-4.62e20 FLOPs, Table 5), CompleteP consistently improves the compute efficiency of deeper models.
>
> We recognize many of these points could be made more clear in the paper and will revise the results and limitations sections to better highlight this context and clarify how our results substantiate these claims. Thank you for helping to improve our submission!
>
> ```
> While compute optimal settings provide good insights, modern day small LLMs are trained far beyond the 20TPP number. Impact of data can be seen from Figure 2 bottom to Figure 3. This makes the claim a little harder to verify. Will the benefits persist under different training regimes (e.g., longer training, different batch size schedules)?
> ```
> This is a great point: we also believe that the practical utility of a parameterization such as CompleteP rests in large part on its usefulness across a range of models, datasets, and training procedures. While it is not computationally feasible to exhaustively test $\alpha = 1$ across as many scenarios as we'd like, we do believe our work covers at least the settings below. We will emphasize them in the introduction of our revision:
> - **Varying token counts.** We demonstrate depth HP transfer for CompleteP at 300M tokens (most models under-trained) and 20 TPP (Figures 2 and 3). We also compare the performance of parameterizations across N:L ratios and show CompleteP is superior to $\mu$P and $\alpha=0.5$ for both 20 TPP (Figure 4 and Table 2) and 200 TPP (Table 3) settings. We believe this covers a reasonable range of practical scenarios, ranging from under-trained to compute-optimal to over-trained. We now see that the 200 TPP results (Table 3) are buried in the appendix with no references. We will move these to the main body.
> - **Varying batch-size prescriptions.** We demonstrate depth HP transfer for CompleteP both at constant batch size (Figure 2) and with batch size as a function of training FLOPs (Equation 47) to enhance compute-efficiency (Figure 3). Unfortunately we did not test *batch size schedules* in which batch size varies across optimization steps. This is one of the many configurations we would have liked to test but didn't have the compute to prioritize.
>
> ```
> Modern day LLMs are moving towards longer context, MoEs, Group Query Attention, RMSNorm and other enhancements to architecture. Would these analysis persists for these architectures too?
> ```
> This is a really good question! We can definitely do a better job at explaining the generalizability of the CompleteP scaling for practitioners, as well as explain the math for how to derive other cases. There are roughly three separate categories of generalizability:
>
> 1. No modifications required. This includes LayerNorm at any position (pre, post, QK-norm etc.), adding biases at different locations, LR schedule, and other well-normalized Adam-like algorithms (e.g. SignGD, Adagrad, Lion), or position embeddings (learned, RoPE, ALiBi, NoPE, etc.). The reason is that these do not change the scale of both the forward pass and hidden layer updates due to training, with respect to width and depth. LayerNorm, for example, maintains the $\Theta(1)$ scale of each neuron, and as long as the update maintains the $\Theta(1)$ scale, then no changes are required.
>
> 2. Slight modifications are required and known. A good example of this is SGD, where the learning rate scaling is derived in Table 1 of Bordelon et al. [8] (up to an equivalent ABC-reparameterization). Notice here that compared to Adam, the learning rate of SGD needs to be scaled up due to the prefactors in front of both the weight matrices and the residual branch, where as these extra factors are canceled out in Adam due to normalization.
>
> 3. More theoretical derivations required. This includes MoE, long context, batch size (and schedule), gradient clipping, LAMB, and momentum. Some of these are a relatively straight forward exercise to derive (e.g. LAMB and other optimization algorithms), others are less straight forward. In particular, any scaling that requires increasing the number of data points and training steps along with width and depth remains theoretically challenging. Our approach in this work computes the theoretical scaling for finitely many data points and training steps, and testing whether or not hyperparameter transfer empirically, which has been yielding very good results (see e.g. Figure 3).
>
> To improve the paper, we will include a more comprehensive and pedagogical section on generalizability, so can improve the accessibility of adopting CompleteP in a variety of training scenarios. Thank you for raising these concerns!
>
> ```
> Can the authors provide guidance on hyperparameter sensitivity analysis for practitioners for CompleteP?
> ```
> This is a great question. First we note that Figures 2, 3, and 8 provide a hyperparameter sensitivity analysis for learning rate, initialization, and AdamW $\epsilon$. However, ideally we would have tested the sensitivity of more hyperparameters. The theory on which we built CompleteP does not directly give any indication about the sensitivity of optimal HPs. However, we found empirically that *larger TPP leads to less sensitive optimal HP configurations* (see Figure 2 vs 3), and so that would be our main piece of advise to practitioners: it is likely that more over-trained models have more robust HPs. We didn't think to specifically formulate this point and will include it more prominently in our revision.

---

### Official Review · Reviewer_skfW · 2025-07-02

**Clarity:** 4
**Significance:** 4
**Originality:** 2
**Rating:** 5
**Confidence:** 4

**Summary:**

This work studies depth-dependent parameterizations for hyperparameter transfer when training deeper models. The authors introduce the parameterization completeP, which they show guarantees non-lazy learning at each layer as models become deeper. They demonstrate that this parameterization enables effective hyperparameter transfer as model depth increases and improves performance relative to standard parameterization, μP, and another depth-dependent parameterization (i.e., α=0.5). They additionally demonstrate that completeP allows for a wider range of compute-efficient width/depth ratios. Their results are shown on large language models trained over a sweep of model sizes using large datasets.

**Questions:**

# Comments

- I don’t understand what you are plotting in Figure 1(c) — the colored areas represent the “range of efficient N:L,” but how do you define this? I suggest you describe it clearly in the caption or within the figure itself.

- Lines 35–37 don’t make sense to me as written. You introduce that the depth-dependent rescaling factor is governed by a single parameter α ∈ [0.5, 1], define α=1 as your algorithm completeP, and then state “Since α=0.5 and α=1 are the two most promising potential candidates for depth scaling...” but it’s unclear why α=0.5 is included. Has α=0.5 been previously defined in the literature? I think adding a sentence or two explaining what prior work has done with α=0.5 and how your work extends this will make the introduction clearer.

- Since you continuously compare to the α=0.5 setting from the work “Feature Learning in Infinite-Depth Neural Networks,” I would suggest giving this setting a name.

- I think you should also briefly discuss what would happen for α ∈ (0.5, 1.0), even though I agree with your logic that the boundary cases are the most important.

- In Kaplan et al., they use a version of the WebText dataset, while your work uses SlimPajama, correct? It’s worth explicitly highlighting this difference.

**Ethical Concerns:**

["NO or VERY MINOR ethics concerns only"]

**Final Justification:**

This is a strong paper with impressive empirical results and I recommend acceptance. While, not the most original, it is very well executed and the rebuttal helped clarify any remaining questions I had.

**Limitations:**

yes

**Paper Formatting Concerns:**

No issues

**Quality:**

4

**Strengths And Weaknesses:**

# Strengths

- Tackles an extremely important problem for scaling large language models and improving the performance of state-of-the-art systems.
- Presents an impressive display of empirical testing.
- High-quality experiments, well written, and significant work overall.

# Weaknesses

- This work is not particularly original; it primarily extends prior ideas (references 6–8), but does an excellent job empirically testing and validating these results to demonstrate that completeP is the best setting. That said, I think this work could do a better job distinguishing its contributions from prior works (see my comments below)

- Despite the impressive empirical sweep, there are still limitations. For example, as discussed in lines 79–88, there exist other layer-dependent rescalings that are not considered in this empirical study, and the results are only performed on the single dataset SlimPajama. I suggest moving the discussion of these limitations, such as the one you include in Appendix B, up into the main conclusion section.

---

> ### Author Rebuttal · Authors · 2025-07-31
>
> Thank you for the detailed feedback and your positive notes regarding the paper's impact, impressive empirical results, and clear presentation.
>
> ```
> Despite the impressive empirical sweep, there are still limitations. For example, as discussed in lines 79–88, there exist other layer-dependent rescalings that are not considered in this empirical study, and the results are only performed on the single dataset SlimPajama. I suggest moving the discussion of these limitations, such as the one you include in Appendix B, up into the main conclusion section.
> ```
> This is a great suggestion! We will integrate the limitations section (Appendix B) into the main body and update it to highlight that we don't empirically test FLERM [49] or Modula [50] and update lines 629-630 to say "Our empirical results only test the setting of next-token prediction decoder-only pre-LN transformers trained with AdamW on the SlimPajama text dataset."
>
> ```
> I don’t understand what you are plotting in Figure 1(c) — the colored areas represent the “range of efficient N:L,” but how do you define this? I suggest you describe it clearly in the caption or within the figure itself.
> ```
> The colored areas in Figure 1-right (and Figure 4f) represent the N:L range ensures <= 1\% loss increase relative to the most compute-efficient point collected. The process for determining the colored areas in Figure 1-right is outlined on lines 817-819 and in Figure 13. On line 160 we do state "See Appendix I.3 for extensive details on model shapes and plots" but this seems woefully inadequate given that we include this plot on the first page of the paper. To remedy this we will update the manuscript to better explain this methodology near where the figure appears and have more informative captions. Thank you for noting this was unclear!
>
> We will also clarify Figure 1 (middle) by changing the Y-axis label to "FLOP savings vs. muP *at same depth*" and noting the black points are "Compute-optimal *(across all depths)*".
>
> ```
> Lines 35–37 don’t make sense to me as written. You introduce that the depth-dependent rescaling factor is governed by a single parameter $\alpha \in [0.5, 1]$, define $\alpha=1$ as your algorithm completeP, and then state “Since $\alpha=0.5$ and $\alpha=1$ are the two most promising potential candidates for depth scaling...” but it’s unclear why $\alpha=0.5$ is included. Has $\alpha=0.5$ been previously defined in the literature? I think adding a sentence or two explaining what prior work has done with $\alpha=0.5$ and how your work extends this will make the introduction clearer.
> ```
> Thank you for pointing out the unclear reference to $\alpha=0.5$ in the introduction. On lines 72-76 we briefly mention how [7] argued $\alpha=0.5$ was the best in practice and this paper shows CompleteP ($\alpha=1$) is superior to $\alpha=0.5$. However this information is not presented in the introduction, making the choice of $\alpha=0.5$ on lines 35-37 unmotivated and weakening the introduction by assuming too much prior knowledge. We will update the introduction to provide more background on why both $\alpha=0.5$ and $\alpha=1$ are promising candidates for depth scaling.
>
> ```
> Since you continuously compare to the $\alpha=0.5$ setting from the work “Feature Learning in Infinite-Depth Neural Networks,” I would suggest giving this setting a name.
> ```
> Thank you for the suggestion. This is something we discussed while writing but ultimately decided against it. We considered naming the $\alpha=0.5$ setting "LazyP" because it will be in the lazy learning regime in the infinite-depth limit. However this is not a unique name since any $\alpha<1$ could also be called "LazyP". On the other hand, CompleteP is a unique name because only $\alpha=1$ ensures complete feature learning. We were also concerned that "LazyP" would be too demeaning to the $\alpha=0.5$ setting. We hope this clarifies our thinking!
>
> ```
> I think you should also briefly discuss what would happen for $\alpha \in (0.5, 1)$, even though I agree with your logic that the boundary cases are the most important.
> ```
> Yes we agree and include a brief discussion in Section 6 (lines 271-275). We see now that some discussion of the $\alpha \in (0.5, 1)$ case should also be included in the introduction and related work to help motivate why we chose to study these specific boundary cases. Thank you for raising this point!
>
> ```
> In Kaplan et al., they use a version of the WebText dataset, while your work uses SlimPajama, correct? It’s worth explicitly highlighting this difference.
> ```
> Good point. SlimPajama is a mix of web text (Common Crawl and C4), academic prose (Books, arXiv, and Wikipedia), and code (Github and Stack Exchange). Training on a more diverse and high quality dataset modernizes our setup and potentially makes our results more applicable to contemporary LLM training. We should also mention that Kaplan et al. [2] use the standard parameterization in their experiments. We will highlight this difference within lines 154-160. Thank you for the constructive feedback!

---

> > ### Comment · Reviewer_skfW · 2025-08-04
> >
> > Thank you for the clarifications. My questions have been fully addressed. I stand by my evaluation that this is a strong contribution to the field and well deserving of acceptance.

---

### Official Review · Reviewer_yU41 · 2025-07-03

**Clarity:** 3
**Significance:** 3
**Originality:** 2
**Rating:** 5
**Confidence:** 2

**Summary:**

Hyperparameter (HP) tuning is expensive at scale. Prior work like µP has shown how to enable zero-shot HP transfer when scaling model width. This work extends that line of research by providing a recipe for scaling HPs—such as bias learning rates, weight decay, Adam epsilon, and LayerNorm parameters, enabling a zero-shot HP transfer from shallow to deep networks. They show up to 34% FLOP savings.

**Questions:**

1. The authors demonstrate both empirically and theoretically that setting the residual scaling α = 1 enables full, non-lazy feature learning across depth. How does this finding relate to, or differ from, the s=1 regime described in the prior theoretical work Meta-Principled Family of Hyperparameter Scaling Strategies, ref. [41]? In what way is Desideratum 3 novel beyond what is implied by the s=1 maximal-update regime?

2. There are many hyperparameters that influence training dynamics as model size increases. This work focuses on scaling a specific subset: bias learning rates, weight decay, Adam epsilon, and LayerNorm parameters. Could the authors elaborate on why these were prioritized over others—such as batch size, learning rate warmup schedules, gradient clipping, or optimizer momentum—and whether the proposed method is sensitive to those remaining hyperparameters?

3. The work is developed and evaluated using pre-LayerNorm Transformer architectures, which are commonly used for deep models due to their favorable gradient dynamics. Could the authors comment on whether CompleteP generalizes to alternative normalization schemes, such as post-LN or sandwich-LN architectures? Would any modifications to the scaling rules (e.g., for residuals or LayerNorm) be required in those settings?

4. The paper demonstrates the proposed scaling strategy using AdamW, but does not explore its interaction with other optimizers such as SGD, LAMB, or Lion. Could the authors comment on whether CompleteP generalizes to these optimizers, and whether additional tuning or scaling adjustments would be required?

5. In Figure 4 (d), it appears that the scaling law fit is based on only three data points. Could the authors clarify whether this is the case, and if so, how robust the fit is to such limited data? Additionally, for Figure 4(f), how are the shaded region estimated?

**Ethical Concerns:**

["NO or VERY MINOR ethics concerns only"]

**Final Justification:**

This is a technically solid paper, with high impact for AI practitioners. I maintain my accept score.

**Limitations:**

yes

**Quality:**

3

**Strengths And Weaknesses:**

(+) practical recipe for training deep transformers

(+) enabling zero-shot HP transfer, reducing HP tuning budget

(+) unlocking model shapes better suited for different hardware

(+) simple implementation, require only a few lines of code


(-) lack of direct theoretical comparison to prior work

(-) selective hyperparameter focus, the method only address a subset of HPs

(-) unclear if proposed approach generalize beyond transformer and Adam optimizer

---

> ### Author Rebuttal · Authors · 2025-07-31
>
> Thank you for the detailed and constructive feedback, and your positive notes recognizing CompleteP as a simple and practical method for training deep transformers which enables zero-shot hyperparameter transfer and unlocks hardware-efficient model shapes.
>
> ```
> Does CompleteP generalize across batch size, learning rate warmup schedules, gradient clipping, optimizer momentum, post-LN, sandwich-LN, SGD, LAMB, or Lion? Would additional tuning or scaling adjustments be required?
> ```
> This is an excellent question relevant for many practitioners! We can definitely do a better job at explaining the generalizability of the CompleteP scaling, as well as explain the math for how to derive other cases. There are roughly three separate categories of generalizability:
>
> 1. No modifications required. This includes LayerNorm at any position (pre, post, QK-norm etc.), adding biases at different locations, LR schedule, and other well-normalized Adam-like algorithms (e.g. SignGD, Adagrad, Lion), or position embeddings (learned, RoPE, ALiBi, NoPE, etc.). The reason is that these do not change the scale of both the forward pass and hidden layer updates due to training, with respect to width and depth. LayerNorm, for example, maintains the $\Theta(1)$ scale of each neuron, and as long as the update maintains the $\Theta(1)$ scale, then no changes are required.
>
> 2. Slight modifications are required and known. A good example of this is SGD, where the learning rate scaling is derived in Table 1 of Bordelon et al. [8] (up to an equivalent ABC-reparameterization). Notice here that compared to Adam, the learning rate of SGD needs to be scaled up due to the prefactors in front of both the weight matrices and the residual branch, where as these extra factors are canceled out in Adam due to normalization.
>
> 3. More theoretical derivations required. This includes MoE, long context, batch size (and schedule), gradient clipping, LAMB, and momentum. Some of these are a relatively straight forward exercise to derive (e.g. LAMB and other optimization algorithms), others are less straight forward. In particular, any scaling that requires increasing the number of data points and training steps along with width and depth remains theoretically challenging. Our approach in this work computes the theoretical scaling for finitely many data points and training steps, and testing whether or not hyperparameter transfer empirically, which has been yielding very good results (see e.g. Figure 3).
>
> To improve the paper, we will include a more comprehensive and pedagogical section on generalizability, so can improve the accessibility of adopting CompleteP in a variety of training scenarios. Thank you for raising these concerns!
>
> ```
> Lack of direct theoretical comparison to prior work. The authors demonstrate both empirically and theoretically that setting the residual scaling $\alpha=1$ enables full, non-lazy feature learning across depth. How does this finding relate to, or differ from, the $s=1$ regime described in the prior theoretical work Meta-Principled Family of Hyperparameter Scaling Strategies, ref. [41]? In what way is Desideratum 3 novel beyond what is implied by the $s=1$ maximal-update regime?
> ```
> Thank you for pointing out the connection of our work to Yaida [41]. The main contribution of Yaida [41] is identifying how to have consistent training dynamics (and hence HP transfer) as a function of a parameter $s\in [0,1]$ in joint limits of diverging depth $L$ and width $n$ when fixing the ratio of $\gamma = \frac{L}{n^{1-s}}$ in MLPs. Since the roles of depth in MLPs and ResNets are quite different, it is hard to directly compare our parameterization to his. Moreover, since Yaida [41] considered non-shaped activations, he found that in the $\mu P, s=1$ regime, one must keep the resulting ratio $\gamma = L$ constant to have stable training. In contrast, we find that one can have diverging $n,L$ separately and still have HP transfer and stable training dynamics in ResNets.
>
> We will add this discussion to the manuscript as this is an interesting and relevant connection to our work.
>
> ```
> In Figure 4 (d), it appears that the scaling law fit is based on only three data points. Could the authors clarify whether this is the case, and if so, how robust the fit is to such limited data?
> ```
> Yes the scaling law fit in Figure 4d is based on three data points. We acknowledge the limitations of this approach on lines 633-637.
>
> However each of three points are the most compute-efficient point from each group of points where we sweep 7-10 values of N:L for each parameter count, so they are likely to be an accurate picture of the compute-efficient frontier. The three points also span a 256x range of training FLOPs (1.56e18-3.99e20). Table 5 provides further details on this experimental design. We mainly use the scaling law fits for interpolation rather than a heavy extrapolation regime where fitting to limited data could make predictions unreliable.
>
> We believe the above discussion provides important context so we will update the limitations section to include it. Thank you for raising this concern!
>
> ```
> For Figure 4(f), how are the shaded regions estimated?
> ```
> The shaded regions in Figure 4f (and Figure 1 right) represent the N:L range ensures <= 1\% loss increase relative to the most compute-efficient point collected. The process for determining the shaded regions in Figure 4f is outlined on lines 817-819 and in Figure 13. On line 160 we do state "See Appendix I.3 for extensive details on model shapes and plots" but this seems woefully inadequate given that we include this plot on the first page of the paper. To remedy this we will update the manuscript to better explain this methodology near where the figure appears and have more informative captions. Thank you for noting this was unclear!
>
> We will also clarify Figure 1 (middle) by changing the Y-axis label to "FLOP savings vs. muP *at same depth*" and noting the black points are "Compute-optimal *(across all depths)*".

---

> ### Comment · Reviewer_yU41 · 2025-08-01
>
> Thank you for the detailed response! My concerns have been adequately addressed, and I have no further questions. I agree that including a section on generalizability would further strengthen the paper’s value for practitioners.

---

### Note · Authors · 2025-08-12

We would like to thank all our reviewers for their thoughtful feedback that has helped strengthen this work!

---

### Decision · Program_Chairs · 2025-09-17

**Decision:**

Accept (poster)

**Comment:**

This submission proposes a parameterization for training neural networks that avoids the lazy regime (where features at intermediate layers of the network don't evolve during training) while still ensuring hyperparameter transfer across depth and width scaling (e.g., we can find the good stepsize with a small, cheap model and then use this on the big, expensive model).

Reviewers received the paper warmly, with all of them giving mostly positive initial reviews. They highlighted the paper's strengths as its rigorous approach, its practicality (few lines of code for implementation, easy HP tuning via transfer), and its comprehensive numerical experiments, which primarily cover language modeling.

The paper could have been improved by looking at optimization algorithms beyond Adam or by being more innovative/original in its theoretical aspects, since the parameterization ultimately used is quite closely related to what has already been proposed in some other works on parameterizations for hyperparameter transfer. I find that these flaws are not substantial enough to warrant rejection since the parameterization and study involved are novel, even if the idea that parameterizations can be chosen to ensure hyperparameter transfer was not novel per se.

During the rebuttal period, the authors addressed almost every concern brought up by the reviewers, with some reviewers improving their score as a result, and ultimately ended with all positive reviews. I agree with the reviewers that this paper should be accepted and that the results contained within are of great interest to the NeurIPS community.